# SORA: FREE SECOND ORDER ATTACKS IN FAST ADVERSARIAL TRAINING

## ABSTRACT

Adversarial Training (AT) is a leading defense against adversarial examples but often suffers from *Catastrophic Overfitting* (CO) in efficient single-step variants, where robustness to multi-step attacks collapses despite high single-step performance. We address this failure mode with two contributions. First, we identify *Epsilon Overfitting* (EO), a previously overlooked phenomenon in which fixed perturbation magnitudes exacerbate CO, and show that introducing perturbation variability significantly improves robust generalization across different architectures and datasets. Second, we propose **PertAlign** (Perturbation Alignment), a theoretically grounded, computationally negligible metric that predicts CO onset by measuring gradient alignment across attack stages. Leveraging these insights, we introduce **SORA**, an adaptive step-size adversarial training method that dynamically adjusts perturbations based on loss-surface geometry. SORA consistently prevents CO, achieves state-of-the-art robustness and clean accuracy, and generalizes across datasets and architectures using a single fixed set of hyperparameters. Extensive experiments on diverse datasets and architectures, show that SORA matches or surpasses the robustness of prior methods while delivering higher clean accuracy and superior efficiency. Code is available at https://anonymous.4open.science/r/2026_ICLR_SORA.

## 1 INTRODUCTION

Deep neural networks have achieved remarkable success across a wide range of domains. However, their vulnerability to adversarial examples poses significant challenges to their reliability and security (Szegedy et al., 2014). These adversarial examples are often subtle and imperceptible to the human eye, yet can cause models to make incorrect predictions with high confidence.

Adversarial Training (AT) (Madry et al., 2019) has emerged as one of the most effective defenses against adversarial examples (Athalye et al., 2018). By training models on adversarially perturbed samples, AT improves robustness to attacks during deployment. Despite these benefits, multi-step AT methods such as Projected Gradient Descent (PGD) (Madry et al., 2019) are computationally expensive, as they require several gradient ascent steps to generate adversarial examples at each training iteration. This cost severely limits their scalability to large datasets and deep architectures.

To mitigate this overhead, single-step adversarial training approaches (Shafahi et al., 2019; Wong et al., 2020) have been proposed. While significantly more efficient, these methods introduce new challenges. Notably, Wong et al. (2020) identified a failure mode in single-step AT, commonly observed when using the Fast Gradient Sign Method (FGSM) (Goodfellow et al., 2015), that has been termed *Catastrophic Overfitting* (CO); related limitations of one-step attacks were also discussed in earlier AT work (Madry et al., 2019). Here, robust accuracy against multi-step attacks such as PGD collapses to nearly $0\%$, while FGSM accuracy can rise sharply and even exceed the clean accuracy. This suggests that the model overfits to the specific attack used in training and becomes brittle against stronger iterative attacks.

Although many methods have been proposed to address this challenge, we argue that an effective solution to Catastrophic Overfitting (CO) should satisfy the following properties:

- **Robustness across datasets and architectures.** It should reliably prevent CO on a diverse range of datasets and models. As shown in Section 6 and Appendix H, our proposed

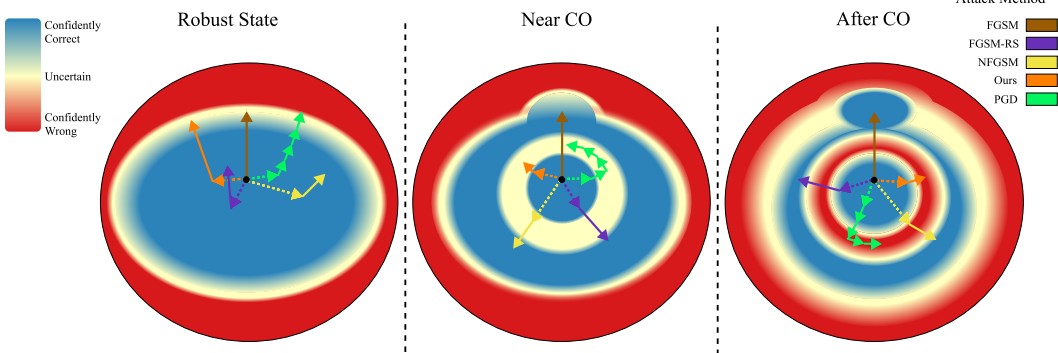

Figure 1: Evolution of the loss landscape geometry around a sample during FGSM training. **Left:** Early in training, before CO, the model is robust, with FGSM and PGD accuracies closely matched. The loss surface is approximately semi-linear. **Middle:** A few batch updates before CO, the decision boundary begins to wrap around the original and adversarial examples, forming a nonlinear region that is not yet misclassified. **Right:** This nonlinearity escalates, leading to misclassification in that region, which stronger attacks such as PGD exploit. Perturbations from other attacks and our method (SORA) are also visualized.

method, Second Order Adaptive (SORA), succeeds not only on commonly used datasets but also on challenging benchmarks where existing single-step techniques often fail.

- **High clean and robust accuracy with low cost.** A known trade-off exists between clean and robust accuracy (Tsipras et al., 2019; Zhang et al., 2019). Many prior methods achieve strong robustness by restricting model flexibility (e.g., stronger regularization or reduced capacity), which harms clean accuracy. In contrast, SORA maintains high clean accuracy and SOTA robustness while adding negligible computational and memory overhead by efficiently choosing the attack step size distribution.

- **Dataset and architecture agnostic hyperparameters.** Several methods require extensive, dataset- and model-specific hyperparameter tuning to avoid CO, which undermines the original goal of Fast Adversarial Training (FAT): robustness with minimal training cost. In our experiments (Section 6 and Appendix H), we fix SORA's hyperparameters across all settings and adopt the single best configuration reported in prior work for a fair comparison, demonstrating strong generalization without costly tuning for each setting.

To the best of our knowledge, no metric has been explicitly designed for predicting CO without significant cost. Nevertheless, several existing measures can serve as indirect early indicators of CO onset (Andriushchenko & Flammarion, 2020; Lin et al., 2024; Rocamora et al., 2024). These indicators often incur high computational cost, requiring extra backpropagation or forward passes. To address this gap, we propose PertAlign, a fast, gradient-based metric that is theoretically linked to the curvature of the loss landscape and can predict CO with negligible overhead. It uses gradients from the first and second attack iterations. We further show that PertAlign can anticipate CO earlier than existing indicators.

In addition, we identify a previously overlooked phenomenon, which we term *Epsilon Overfitting* (Figure 1). We show in Section 3 that variability in the attack step size plays a crucial role in the emergence of CO. By incorporating this insight, SORA achieves improved robustness across a wider range of settings compared to previous baselines. Our experiments demonstrate that methods incorporating perturbation variability generalize more effectively and consistently avoid CO.

**Our contributions are as follows:**

- We introduce and analyze *Epsilon Overfitting*, demonstrating its importance for robust generalization and CO.

- We propose *PertAlign*, a theoretically grounded and computationally efficient metric for early and reliable CO prediction.

- We present *SORA*, an efficient training paradigm that leverages Epsilon Overfitting and PertAlign to adaptively determine attack step sizes. We show that it matches or surpasses SOTA performance while generalizing across datasets and architectures with minimal hyperparameter tuning.

## 2 BACKGROUND AND RELATED WORK

### 2.1 ADVERSARIAL TRAINING

Adversarial Training (AT) can be formulated as a min–max optimization problem, where the inner maximization seeks adversarial perturbations that maximize the training loss, while the outer minimization updates model parameters to minimize the loss on these adversarial examples (Madry et al., 2019). Formally, this can be expressed as:

$$\min_{\theta} \mathbb{E}_{(x,y)\sim\mathcal{D}} \left[ \max_{\delta\in\Delta} \mathcal{L}\left(f_{\theta}(x+\delta), y\right) \right], \tag{1}$$

where $\theta$ denotes the parameters of the model $f$, $(x, y)$ are the training data drawn from the distribution $\mathcal{D}$, $\mathcal{L}(\cdot)$ denotes the cross-entropy loss function, and $\delta$ is the perturbation confined within a given boundary $\Delta$.

In the standard multi-step AT by Madry et al. (2019), the loss function used is the cross-entropy loss and $\Delta$ is defined as the $\ell_{\infty}$-norm ball with radius $\epsilon$, and the inner maximization is solved via the Projected Gradient Descent (PGD) attack. While effective, this iterative approach significantly increases the computational cost for large-scale models and datasets.

Within AT, a family of methods, collectively referred to as Fast Adversarial Training (FAT), has emerged with the goal of producing robust models at a fraction of the cost of multi-step training. However, while these methods improve efficiency, fast single-step AT approaches are prone to a distinctive failure mode known as Catastrophic Overfitting (CO). Understanding and mitigating CO has thus become a critical research direction.

### 2.2 CATASTROPHIC OVERFITTING

Catastrophic Overfitting (CO) is a phenomenon observed during adversarial training, more prevalently during single-step adversarial training, where robustness against multi-step attacks such as PGD suddenly decreases to 0% just within a few epochs, whereas robustness against the FGSM attack rapidly increases. Many studies have focused on explaining CO and developing FAT variants that avoid it. Broadly, prior works can be grouped by their main mitigation strategy:

**Randomization.** Randomization (noise based) methods inject stochasticity into adversarial example generation to prevent harmful overfitting. Wong et al. (2020) demonstrated that adding random starts to FGSM training can prevent CO. de Jorge et al. (2022) extended this by increasing the magnitude of random noise and removing the clipping step to boost accuracy.

**Regularization.** Regularization-based methods constrain model behavior to discourage CO. Andriushchenko & Flammarion (2020) proposed maximizing gradient alignment within the perturbation set. Lin et al. (2024) identified *abnormal* adversarial examples as a source of CO and mitigated them via loss regularization. Similarly, Rocamora et al. (2024) encouraged local linearity in the loss surface.

**Adaptive Step Sizes.** These methods modify the attack step size based on training dynamics. Nie et al. (2021) used reinforcement learning to determine step sizes. Huang et al. (2022) adapted them using gradient norms.

**Other Approaches.** Some works address CO from alternative perspectives. For instance, Golgooni et al. (2021) omitted updates with negligible gradient magnitudes to stabilize training, while Jia et al. (2022; 2023) use prior-guided adversarial initialization to generate stronger adversarial examples by using high-quality adversarial perturbations from the historical training process.

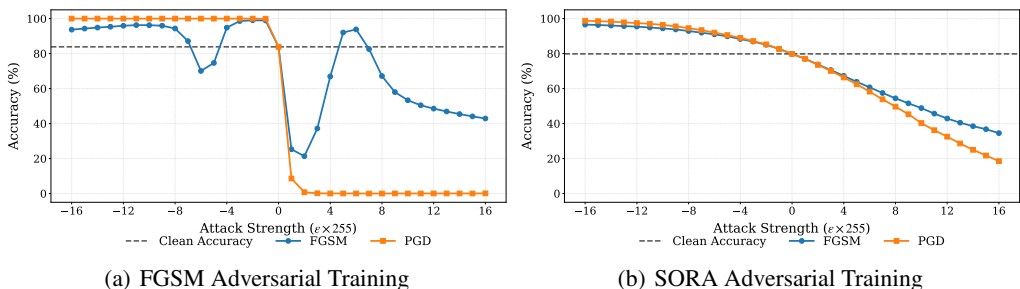

(a) FGSM Adversarial Training        (b) SORA Adversarial Training

Figure 2: Comparison of a model exhibiting CO (subfigure (a)) with a robust model (subfigure (b)), evaluated using FGSM and PGD-10 accuracies across varying $\epsilon$ values.

## 3 EPSILON OVERFITTING

Although numerous studies have examined Catastrophic Overfitting (CO) from various perspectives, a comprehensive understanding of its underlying mechanisms remains elusive. One well-established viewpoint associates CO with changes in the geometry of the model's loss landscape, with several works showing that, at its onset, the loss surface becomes highly distorted (Kim et al., 2020; Kang & Moosavi-Dezfooli, 2021). However, a more precise characterization of this distortion is still lacking. We build on this perspective by analyzing the loss landscape and robustness across different training states, offering a clearer depiction of how the surface evolves throughout CO. Specifically, we investigate whether this distortion follows a consistent, characteristic pattern.

### 3.1 EVOLUTION OF LOSS LANDSCAPE GEOMETRY

A striking property of CO is that it does not degrade clean accuracy, yet it often causes a substantial increase in FGSM accuracy, sometimes even surpassing clean accuracy on the test set. This counter-intuitive effect prompted us to examine FGSM accuracy across a range of $\epsilon$ values for a model trained with a fixed perturbation budget (e.g., $\epsilon = 8/255$), as shown in Figure 2(a).

Two patterns can be observed in this plot. First, there exist certain $\epsilon$ values for which FGSM accuracy drops sharply (e.g., $\epsilon = 2/255$, $\epsilon = {}^{-}6/255$), while for other values it remains unexpectedly high (e.g., $\epsilon = {}^{-}2/255$, $\epsilon = 6/255$). Such high variation in FGSM accuracies indicates pronounced nonlinearity in the loss surface, even along the FGSM perturbation direction used during training. Since the model effectively *overfits* to specific $\epsilon$ values, performing well for those perturbation magnitudes but poorly for others, we refer to this phenomenon as *Epsilon Overfitting* (EO).

EO arises when the attack perturbation magnitude is fixed and relatively large, with limited or no flexibility in magnitude or direction. Under such constraints, the model can learn to reduce the loss only within the narrow regions reached by the fixed perturbation, rather than across the broader $\ell_\infty$-norm ball.

Lin et al. (2024) identified a related phenomenon, *Abnormal Adversarial Examples* (AAEs), adversarial examples with lower loss than the original clean sample. From Figure 1, we see that most AAEs arise in the presence of *Epsilon Overfitting* when adversarial examples are generated with the same step size used during training. In this case, the model enlarges its decision boundary around the adversarial example relative to that around the clean example, producing the observed FGSM accuracy spike, which can even exceed clean accuracy.

### 3.2 OVERCOMING EPSILON OVERFITTING

These findings, along with prior work (Ding et al., 2020; Huang et al., 2022), highlight the importance of adaptive step-size methods. Such methods adjust step sizes in response to local loss curvature, avoiding interpolation over highly nonlinear regions between the clean and adversarial points and instead targeting the local maxima along the perturbation direction.

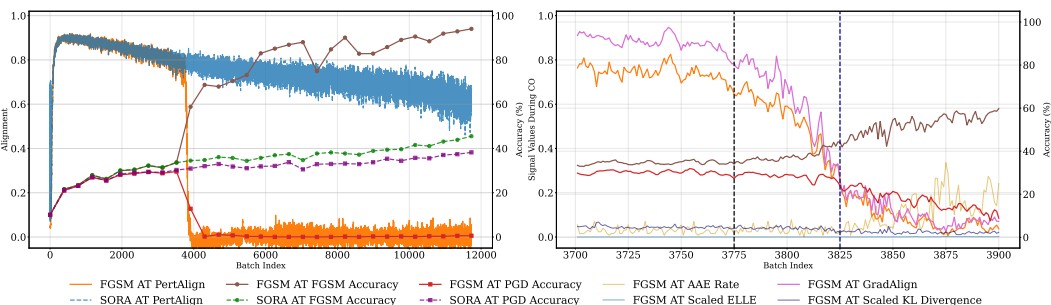

Figure 3: **Left:** Tracking PertAlign during FGSM AT and SORA AT, PertAlign collapses on the occurrence of CO. **Right:** At batch 3775 of FGSM AT, PertAlign and GradAlign begin to drop, forecasting CO, while FGSM and PGD accuracies visibly diverge only around batch 3825. Other metrics, AAE Share of the Batch, Scaled KL Divergence from TRADES, and Scaled ELLE nonlinearity measure, react later in response to the model updates.

The FGSM trend in Figure 2(a), where accuracy evolves in a smooth yet non-monotonic manner for small $\epsilon$, indicates that the loss landscape near the clean point is not purely jagged but exhibits structured curvature. This motivates approximating the neighborhood of $\epsilon = 0$ with a second-order (quadratic) model, thereby enabling curvature-aware step-size selection for loss maximization.

Although second-order methods can explicitly solve the maximization problem via local curvature estimation, their computational cost is prohibitive for large-scale problems due to Hessian computation Ghojogh et al. (2021). Consequently, most practical approaches infer curvature indirectly to strike a balance between cost and performance. In the following section, we present a technique for approximating loss-surface curvature with **nearly zero** additional computational overhead.

## 4 PERTURBATION ALIGNMENT

One of the key characteristics of Catastrophic Overfitting (CO) is that, in most cases, it arises very quickly, often within a single epoch. Once CO occurs, most single-step attack methods cannot recover after the robust accuracy drop, and recovery via multi-step evaluation (e.g., PGD) is computationally expensive. Thus, predicting CO **before** single-step and PGD accuracies diverge can enable training methods to take suitable, low-cost corrective actions. Ideally, we seek an accurate and reliable prognosis tool that can detect when optimization has gone off course, well before the onset of CO.

We propose a novel metric, **PertAlign**, which can predict CO by measuring the non-linearity of the loss surface with negligible additional computational cost. PertAlign is defined as the cosine similarity between:

1. The gradient of the loss with respect to the input, computed during adversarial example generation, and

2. The backpropagation gradient, extended back to the input layer, computed when performing backpropagation on the adversarial example passed through the model.

Formally:

$$\texttt{PertAlign} := \cos\left(\nabla_x \mathcal{L}\big(f_\theta(x'), y\big), \nabla_x \mathcal{L}\big(f_\theta(x' + \delta), y\big)\right), \qquad x' = x + \eta, \qquad (2)$$

where $\eta \sim \mathcal{U}(-\epsilon, +\epsilon)^d$ is the random start, and $\delta = \alpha v$ is a perturbation of scalar step size $\alpha$ in direction $v$. The perturbation direction $v$ is typically chosen as

$$g = \nabla_x \mathcal{L}\big(f_\theta(x'), y\big) \quad \text{or} \quad p = \text{sign}\big(\nabla_x \mathcal{L}\big(f_\theta(x'), y\big)\big).$$

It can be shown that PertAlign captures the local non-linearity of the loss surface (see Appendix A).

**Lemma 4.1.** *Let $f_\theta : \mathbb{R}^d \to \mathbb{R}^{\mathcal{C}}$ be a classifier and let $g = \nabla_x \mathcal{L}(f_\theta(x), y)$, $H = \nabla_x^2 \mathcal{L}(f_\theta(x), y)$. Then for $\alpha \ll 1$, PertAlign can be approximated by:*

$$1 - \text{PertAlign} \approx \frac{\alpha^2}{2} \|h_{\perp g}\|^2 \quad h = \frac{Hv}{\|g\|}$$

*where $h_{\perp g}$ denotes component of $h$ which is orthogonal to $g$.*

More specifically, the misalignment measured by PertAlign correlates with the component of $Hv$ (the Hessian $H$ applied to $v$) that is orthogonal to the gradient. This misalignment is minimized when the gradient is aligned with $Hv$, i.e.,

$$v \parallel H^\dagger g, \tag{3}$$

where $H^\dagger$ is the Moore-Penrose pseudo-inverse of $H$. This condition closely resembles Newton's method and implies that the perturbation direction should align with the second-order optimal update step in the loss landscape (Ghojogh et al., 2021).

In practice, setting $v = p$ yields consistent and reliable behavior, as shown in Figure 3. When CO occurs, PertAlign drops sharply toward zero, indicating that $p$ is no longer aligned with the optimal second-order direction, signaling a failure of the inner maximization step. Conversely, when the loss surface remains approximately linear, PertAlign remains stable rather than collapsing.

As further discussed in Section 3, since at the onset of CO, **before** PGD accuracy declines, the local loss surface becomes highly non-linear and unstable (Andriushchenko & Flammarion, 2020; Kim et al., 2020; Rocamora et al., 2024), by PertAlign directly measuring this non-linearity, it can predict CO earlier than the divergence of single-step and PGD accuracies (Figure 3). Moreover, PertAlign consistently foreshadows CO earlier than other proposed loss-surface linearity metrics.

Note that PertAlign relies only on two quantities: $\nabla_x \mathcal{L}(f_\theta(x), y)$, computed during most common FAT methods, and $\nabla_x \mathcal{L}(f_\theta(x+\delta), y)$, the backpropagation gradient extended by one layer to include the input layer, which is already obtained during the model weight update. As these gradients are part of standard adversarial training, PertAlign incurs no additional computational overhead, making it a fast and reliable metric for CO prediction. The information it provides can also be used to mitigate CO, which our proposed AT method, SORA, leverages in Section 5.

We further examine related approaches, including GradAlign (Andriushchenko & Flammarion, 2020) and ZeroGrad (Golgooni et al., 2021), as detailed in Appendix C. Additional properties of PertAlign, alongside an ablation study, are presented in Appendix A.

## 5 METHODOLOGY

In Section 3, we demonstrated the importance of using an adaptive step size in mitigating both EO and CO. This observation motivates us to explore adaptive step-size strategies more systematically. However, this exploration naturally raises a key question:

*What is the optimal perturbation step size, given the current state of the model?*

When loss maximization is viewed purely as a first-order optimization problem, our options for determining a suitable step size are limited. To address this limitation, we instead analyze single-step loss maximization from a second-order optimization perspective.

It can be shown that, by locally approximating the loss landscape with a quadratic function, the optimal step size for maximizing the loss is given by (see Appendix A for the full derivation):

**Lemma 5.1.** *For a second order loss function, let the perturbation of the input be in the direction of $\text{sign}(g)$:*

$$p = \text{sign}(g) \in \{-1, 0, 1\}^d, \quad v = \alpha p.$$

*Then the optimal step size that maximizes this loss is*

$$\alpha^* = \begin{cases} \min\left(\alpha_{\max}, \frac{\alpha_0}{1 - \frac{p^T g'}{\|g\|_1}}\right), & \frac{p^T g'}{\|g\|_1} < 1, \\ \alpha_{\max}, & otherwise. \end{cases}$$

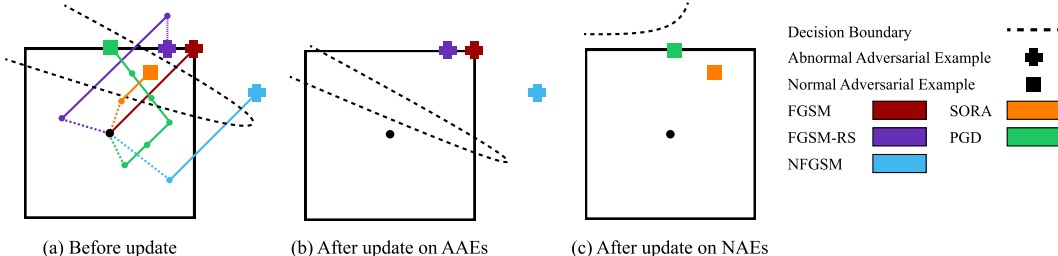

(a) Before update       (b) After update on AAEs       (c) After update on NAEs

Figure 4: When the decision boundary becomes distorted, single-step attacks may produce AAEs, whereas multi-step attacks such as PGD can still reliably generate NAEs. By adapting the attack step size to the local linearity of the loss surface, SORA can also produce NAEs in such scenarios. Training on AAEs tends to exacerbate distortion in the loss surface, while training on NAEs can guide the model toward recovery.

---

**Algorithm 1** **S**econd **Or**der **A**daptive Method (SORA)

1: **Inputs:** # epochs $T$, # batches $M$, radius $\epsilon$, step size $\alpha$, exponential average coefficient $\beta$.
2: **for** Epoch $t = 1, \ldots, T$ **do**
3:      **for** Batch $i = 1, \ldots, M$ **do**
4:          $\boldsymbol{\eta} \sim \mathcal{U}(-\epsilon, \epsilon)^d$                           ▷ Random start
5:          $\boldsymbol{g} \leftarrow \nabla_{\boldsymbol{x}_i} \mathcal{L}\left(\boldsymbol{f}_{\boldsymbol{\theta}}(\boldsymbol{x}_i + \boldsymbol{\eta}), y_i\right)$
6:          $\boldsymbol{\alpha}_i \sim \mathcal{U}\left(0, \alpha^*\right)^d$                   ▷ Element wise step sampling
7:          $\boldsymbol{x}'_i \leftarrow \boldsymbol{x}_i + \boldsymbol{\eta}_i + \boldsymbol{\alpha}_i \odot \mathrm{sign}\left(\boldsymbol{g}\right)$
8:          $\boldsymbol{g}', \boldsymbol{g}_{\boldsymbol{\theta}} \leftarrow \nabla_{[\boldsymbol{x}'_i, \boldsymbol{\theta}]} \mathcal{L}\left(\boldsymbol{f}_{\boldsymbol{\theta}}(\boldsymbol{x}'_i), y_i\right)$        ▷ Backpropagation
9:          $\boldsymbol{\theta} \leftarrow \mathrm{optimizer}(\boldsymbol{\theta}, \boldsymbol{g}_{\boldsymbol{\theta}})$     ▷ Standard parameters update, (e.g. SGD)
10:        Calculate optimal step size for next batch:

$$\alpha^* \leftarrow \begin{cases} \min\left(\alpha_{\max}, \frac{\alpha_0}{1-v}\right), & v < 1, \\ \alpha_{\max}, & \text{otherwise.} \end{cases}$$

11:        $v \leftarrow (1 - \beta) \cdot v + \beta \cdot \frac{\boldsymbol{p}^T \boldsymbol{g}'}{\|\boldsymbol{g}\|_1}$        ▷ Update moving linearity coefficient

---

*where $g' = g + \alpha H p$, $\|\cdot\|_1$ is the $\ell_1$ norm, and $\alpha_{\max}$ is the maximum step size budget, and $\alpha_0$ is a fixed numerator.*

Using this formula as a basis for our attack step size provides a strong initial estimate of an effective perturbation magnitude. To improve stability in cases where the second-order approximation of the loss landscape is inaccurate, and to further mitigate EO, we introduce per-pixel channel diversification: for each channel, we uniformly sample its attack step size from the range $[0, \alpha^*]$. This stochasticity significantly increases the diversity of perturbations, improving both the robustness and universality of the attack, as demonstrated in Section 6. A complete summary of our method is provided in Algorithm 1.

Intuitively, SORA adapts to the evolving state of the model by leveraging the backpropagation gradient from the previous batch. By combining this gradient with the general perturbation direction, it estimates the local non-linearity of the loss surface and adjusts the step size accordingly.

This dynamic adjustment, illustrated in Figure 4, mitigates overshooting, avoids generating AAEs, and prevents the emergence of highly non-linear regions in the loss landscape. The theoretical derivation of the optimal step size along the gradient direction is provided in Appendix A.

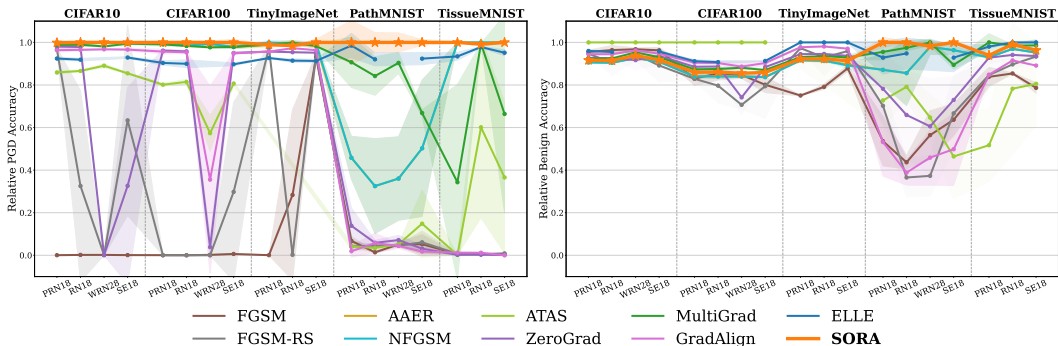

Figure 5: Evaluation of different methods across datasets and architectures. **Left:** SORA attains the highest robust accuracy among single-step methods. **Right:** Corresponding clean accuracy.

## 6 EXPERIMENTS

### 6.1 SETTINGS

**Baselines.** We compare our method against a wide range of single-step Adversarial Training methods, including FGSM-RS (Appendix B.2), GradAlign (Appendix B.3), ZeroGrad and MultiGrad (Appendix B.4), N-FGSM (Appendix B.5), ATAS (Appendix B.6), N-AAER (Appendix B.7), and ELLE (Appendix B.8). For reference, we also evaluate multi-step AT methods, including PGD-10 (Appendix B.9), and TRADES (Appendix B.10), which serve as upper-bound baselines representing idealized performance.

**Datasets and Model Architectures.** We evaluate on multiple architectures, including PreActResNet (Appendix E.2), ResNet (Appendix E.1), and WideResNet (Appendix E.3) from the ResNet family, as well as SENet (Appendix E.4). The training datasets include CIFAR-10 (Appendix D.1), CIFAR-100, (Appendix D.2), and TINYIMAGENET (Appendix D.3). To further evaluate the generalization capability of single-step AT methods, we additionally assess our method and the baselines on the PATHMNIST and TISSUEMNIST from the MEDMNIST collection (Appendix D.4).

**Hyperparameters.** Unless otherwise stated, we set the perturbation budget to $\epsilon = 8/255$ and follow the training setup of Wong et al. (2020), using the SGD optimizer with momentum $0.9$ and weight decay $5 \times 10^{-4}$. For our method we set $\alpha_0 = 0.1$, $\beta = 0.01$, and $\alpha_{\max} = 2\epsilon$. Exact hyperparameter values and training details are included in Appendix B, Appendix E, and Appendix D.

### 6.2 RESULTS

To evaluate the generalizability of single-step adversarial training (SSAT) methods without model- or dataset-specific hyperparameter tuning, we compute each method's PGD-10 and clean accuracy relative to the best-performing method in each setting (Figure 5). A relative accuracy closer to 1 indicates performance closer to the state-of-the-art in that setting. For more details, see Appendix H.

SORA is the only fast method that consistently matches or surpasses the SOTA across all settings, while maintaining higher clean accuracy than competing methods with comparable robust accuracy. This demonstrates a superior trade-off between robustness and clean accuracy.

We report both clean accuracy and robust accuracy for all baselines on the ResNet-18 architecture in Table 1. Robust accuracy is evaluated using AutoAttack (AA) (Croce & Hein, 2020).

Our method matches SOTA performance while maintaining minimal computational and memory overhead (Figure 6). Training time and peak memory usage are measured for all baselines.

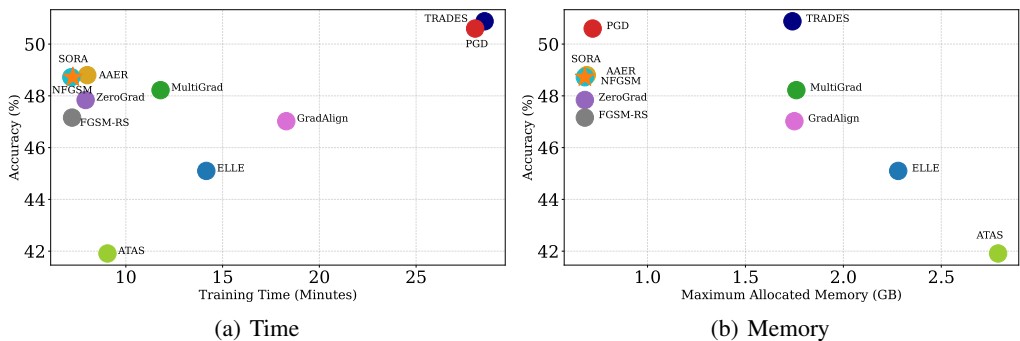

Figure 6: Training time vs. memory usage on CIFAR-10 with PreActResNet-18, trained for 30 epochs, measured on an NVIDIA RTX 4090 GPU. The ★ marks SORA.

Table 1: Training results for ResNet-18 on CIFAR-10, CIFAR-100, and PATHMNIST for $\epsilon = {}^{8}/_{255}$. The best results are shown in **bold** and the second best results have been underlined.

| Method | CIFAR-10 | | CIFAR-100 | | PathMNIST | |
|---|---|---|---|---|---|---|
| | Clean | AutoAttack | Clean | AutoAttack | Clean | AutoAttack |
| FGSM | 84.09 ± 1.85 | 0.00 ± 0.00 | 54.17 ± 1.97 | 0.00 ± 0.00 | 36.69 ± 2.62 | 0.30 ± 0.20 |
| FGSM-RS | 78.90 ± 5.28 | 14.15 ± 20.01 | 50.51 ± 5.11 | 0.00 ± 0.00 | 30.70 ± 7.65 | 1.50 ± 0.41 |
| GradAlign | 82.40 ± 0.22 | 42.49 ± 0.05 | 57.24 ± 0.41 | 20.97 ± 0.17 | 32.62 ± 4.94 | 0.72 ± 1.02 |
| ZeroGrad | 80.74 ± 0.15 | 42.61 ± 0.94 | 56.22 ± 0.51 | 20.87 ± 0.06 | 55.37 ± 17.08 | 1.28 ± 0.72 |
| MultiGrad | 80.66 ± 0.14 | 43.33 ± 0.14 | 55.48 ± 0.49 | 21.63 ± 0.10 | 81.88 ± 1.15 | 29.15 ± 4.48 |
| NFGSM | 78.91 ± 0.30 | 43.96 ± 0.31 | 53.26 ± 0.66 | 21.81 ± 0.06 | 71.90 ± 12.82 | 9.15 ± 6.46 |
| ATAS | **87.23** ± 0.24 | 38.06 ± 0.31 | **63.39** ± 0.16 | 17.50 ± 0.26 | 66.49 ± 4.94 | 0.68 ± 0.34 |
| AAER | 78.91 ± 0.30 | 43.96 ± 0.32 | 53.26 ± 0.66 | 21.92 ± 0.00 | 71.90 ± 12.82 | 9.16 ± 6.48 |
| ELLE | 83.30 ± 0.13 | 40.33 ± 0.13 | 57.53 ± 0.41 | 19.61 ± 0.46 | 79.65 ± 6.09 | 36.22 ± 0.49 |
| SORA (Ours) | 79.81 ± 0.20 | **44.06** ± 0.51 | 54.56 ± 0.18 | **21.99** ± 0.30 | **84.01** ± 0.60 | **38.11** ± 1.94 |
| PGD-10 | **79.17** ± 0.03 | **46.46** ± 0.17 | 53.39 ± 0.40 | **22.93** ± 0.19 | **76.63** ± 1.66 | **48.41** ± 1.31 |
| TRADES | 78.19 ± 0.16 | 46.42 ± 0.16 | **55.00** ± 0.59 | 22.79 ± 0.07 | 18.64 ± 0.00 | 18.64 ± 0.00 |

## 6.3 DISCUSSION ON UNIVERSALITY

Figure 5 reveals a pronounced performance gap between prior methods on both standard datasets (CIFAR-10, CIFAR-100, TINYIMAGENET) and more challenging datasets (PATHMNIST, TISSUEMNIST). This highlights a **lack of generalization** in previous approaches (see Appendix I.2).

Robustness on PATHMNIST and TISSUEMNIST is particularly challenging due to finer image textures, shorter inter-class distances compared to datasets like CIFAR-10 (Figure 8), and class imbalance (Figure 7). Many methods that avoid CO and perform well on standard benchmarks suffer noticeable drops in robust accuracy on these harder datasets. Furthermore, approaches that do maintain robustness, such as ELLE and MultiGrad, incur substantially higher computational or memory costs (Figure 6) or sacrifice clean accuracy such as NFGSM.

## 7 CONCLUSION

We analyzed loss-surface distortion in CO, revealing the critical role of attack perturbation magnitude in **Epsilon Overfitting** (EO) and its close link to CO. Our results demonstrate the strong effectiveness of adaptive step-size strategies in mitigating both phenomena.

We introduced **PertAlign**, a novel, cost-free metric for reliable CO prediction based on loss-surface curvature. Using insights from EO and PertAlign, we developed **SORA**, an adaptive step-size method that adjusts its distribution dynamically according to local linearity. Experiments show that SORA consistently prevents CO, delivers state-of-the-art performance, and generalizes robustly across datasets and architectures.

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

## APPENDIX GUIDE

In Appendix A, we present the **theoretical foundations** of our work.

Details about the baselines, datasets, and architectures are provided in Appendices B, D, and E.

Appendix C discusses related works in greater depth and **expands the theoretical justifications for prior approaches**.

Our ablation studies are presented in Appendix F.

In Appendix G, we offer further insights into how and why PertAlign is effective, and explain why **PertAlign reliably detects catastrophic overfitting**.

Comprehensive results are available in Appendix H. For brevity, a summary is also provided in Figure 5.

We outline our **Research Statement** in Appendix I. Guidelines for reproducing our main results, figures, and ablations are in Appendix I.1. Our approach to **tuning SORA and baselines while ensuring generalizability** is described in Appendix I.2. The **dataset selection process** for reporting performance is detailed in Appendix I.3. Finally, our use of LLMs and the scope of their involvement is documented in Appendix I.4.

## A  THEORETICAL INSIGHTS ON PERTALIGN AND OPTIMAL STEP SIZE

### A.1  OVERVIEW

In this appendix we begin with the general relation in Lemma 4.1, showing how misalignment scales quadratically with step size and depends on the distance between the Hessian-transformed step direction and its projection onto the gradient. Remark A.2 notes that this result is robust to small random noise, while Corollary A.3 specializes it to the gradient direction and Remark A.4 provides an eigenvector-based interpretation. Lemma A.5 then establishes an upper bound on possible misalignment. Proposition A.6 and Proposition A.7 use these findings to derive optimal step size formulas for gradient and sign-gradient directions, respectively. Finally, Lemma A.8 introduces an alternative PertAlign variant.

### A.2  NOTATION

We summarize the notations used throughout this paper:

- $(x, y) \in \mathbb{R}^d \times \{0, 1\}^{\mathcal{C}}$: Input vector and one-hot label, $\mathcal{C}$ classes.
- $f_\theta(\cdot) : \mathbb{R}^d \to \mathbb{R}^{\mathcal{C}}$: Output logits of a neural network classifier parameterized by $\theta$.
- $\mathcal{L}(\cdot, \cdot)$: Cross-entropy loss.
- $D_{\mathrm{KL}}(\cdot \| \cdot)$: Kullback–Leibler (KL) divergence.
- $\|\cdot\|$: Euclidean norm $\ell_2$ unless otherwise stated.
- $\cos(a, b)$: Cosine similarity between vectors $a$ and $b$ defined as $\frac{a^T b}{\|a\| \cdot \|b\|}$.
- $g = \nabla_x \mathcal{L}(f_\theta(x), y)$: Gradient of the loss w.r.t. input $x$.
- $p = \mathrm{sign}\left(\nabla_x \mathcal{L}(f_\theta(x), y)\right)$: Sign of the gradient of the loss w.r.t. input $x$.
- $H = \nabla_x^2 \mathcal{L}(f_\theta(x), y)$: Hessian matrix.
- $\mathcal{U}(a, b)$: A uniform distribution on $[a, b]$.
- $a \odot b$: The Hadamard product between vectors $a$ and $b$.
- $\eta \sim \mathcal{U}(-k\epsilon, +k\epsilon)$: Denotes random noise.
- $\delta$: Input perturbation.
- $\Pi(\cdot)$: Projection operator.

### A.3  PERTALIGN–HESSIAN CORRELATION

**Lemma A.1** (PertAlign–Hessian relationship for arbitrary step direction $v$). *Let $f_\theta : \mathbb{R}^d \to \mathbb{R}^{\mathcal{C}}$ be a classifier and let $g = \nabla_x \mathcal{L}(f_\theta(x), y)$, $H = \nabla_x^2 \mathcal{L}(f_\theta(x), y)$. Then for $\alpha \ll 1$,* PertAlign *can be approximated by:*

$$1 - \text{PertAlign} \approx \frac{\alpha^2}{2} \|h_{\perp g}\|^2 \quad h = \frac{Hv}{\|g\|}$$

*where $h_{\perp g}$ denotes component of $h$ which is orthogonal to $g$.*

*Proof.* **Note that in practice, in cases where $v$ is computed from $x$ (e.g. $v = g$ or $v = p$), we detach $v$ from the PyTorch computation graph after calculating its value. As a result, its functionality as a function of $x$ is removed, and in the Taylor expansion shown below we can assume $v$ is fixed with respect to $x$.**
For the sake of brevity we let $L(x) := \mathcal{L}(f_\theta(x), y)$. Using a second-order Taylor expansion,

$$\nabla_x L(x + \delta) = \nabla_x L(x) + H\delta + \mathcal{O}(\|\delta\|^2) = g + \alpha Hv + R, \quad R = \mathcal{O}(\alpha^2).$$

Let $g' := \nabla_x L(x + \delta)$, then:

$$g^T g' = \|g\|^2 + \alpha g^T Hv + g^T R, \tag{4}$$

$$\|g'\|^2 = \|g\|^2 + 2\alpha g^T Hv + \alpha^2 \|Hv\|^2 + 2g^T R + \mathcal{O}(\alpha^3). \tag{5}$$

Using the cosine formula from equation 2:

$$\text{PertAlign} = \frac{g^T g'}{\|g\| \cdot \|g'\|}.$$

Substituting equation 4 and equation 5 and defining

$$k := \frac{g^T H v}{\|g\|^2}, \quad \gamma := \frac{g^T R}{\|g\|^2},$$

we obtain

$$\text{PertAlign} = \frac{1 + \alpha k + \gamma}{\sqrt{1 + 2\alpha k + 2\gamma + \alpha^2 \frac{\|Hv\|^2}{\|g\|^2} + \mathcal{O}(\alpha^3)}}.$$

Applying the expansion $\sqrt{1+\epsilon} \approx 1 + \frac{\epsilon}{2} - \frac{\epsilon^2}{8}$ and setting $\epsilon = 2\alpha k + 2\gamma + \alpha^2 \frac{\|Hv\|^2}{\|g\|^2} + \mathcal{O}(\alpha^3)$, we get

$$\text{PertAlign} = \frac{1 + \alpha k + \gamma}{1 + \alpha k + \gamma + \alpha^2 \frac{\|Hv\|^2}{2\|g\|^2} - \frac{\alpha^2 k^2}{2} + \mathcal{O}(\alpha^3)}.$$

Let

$$\epsilon_1 = \alpha k + \gamma, \qquad \epsilon_2 = \alpha k + \gamma + \alpha^2 \frac{\|Hv\|^2}{2\|g\|^2} - \frac{\alpha^2 k^2}{2} + \mathcal{O}(\alpha^3)$$

Then,

$$\frac{1 + \epsilon_1}{1 + \epsilon_2} = (1 + \epsilon_1)\left(1 - \epsilon_2 + \epsilon_2^2 + \mathcal{O}(\epsilon_2^3)\right)$$

$$= 1 + \epsilon_1 - \epsilon_2 - \epsilon_1\epsilon_2 + \epsilon_2^2 + \mathcal{O}(\alpha^3)$$

$$= 1 + \alpha k + \gamma - \alpha k - \gamma - \alpha^2 \frac{\|Hv\|^2}{2\|g\|^2} + \frac{\alpha^2 k^2}{2} - \alpha^2 k^2 + \alpha^2 k^2 + \mathcal{O}(\alpha^3)$$

$$= 1 - \alpha^2 \frac{\|Hv\|^2}{2\|g\|^2} + \frac{\alpha^2 k^2}{2} + \mathcal{O}(\alpha^3)$$

Thus,

$$\text{PertAlign} = 1 - \frac{\alpha^2}{2}\left(\frac{\|Hv\|^2}{\|g\|^2} - k^2\right) + \mathcal{O}(\alpha^3) \tag{6}$$

We can rewrite equation 6 in another form:

$$\frac{\|Hv\|^2}{\|g\|^2} - k^2 = \frac{(Hv)^T(Hv) - (kg)^T(kg)}{\|g\|^2}$$

$$= \frac{(Hv)^T(Hv) - 2k(kg^T g) + (kg)^T(kg)}{\|g\|^2}$$

$$= \frac{(Hv)^T(Hv) - 2kg^T H^T v + (kg)^T(kg)}{\|g\|^2}$$

In the last equality, we used the symmetry of the Hessian matrix.

$$\frac{(Hv)^T(Hv) - 2(Hv)^T(kg) + (kg)^T(kg)}{\|g\|^2} = \frac{(Hv - kg)^T(Hv - kg)}{\|g\|^2} = \frac{\|Hv - kg\|^2}{\|g\|^2}$$

Which leads us to

$$\text{PertAlign} = 1 - \frac{\alpha^2}{2}\left\|\frac{Hv}{\|g\|} - \frac{kg}{\|g\|}\right\|^2 + \mathcal{O}(\alpha^3)$$

$$= 1 - \frac{\alpha^2}{2}\left\|\frac{Hv}{\|g\|} - \frac{g^T Hvg}{\|g\|^2}\right\|^2 + \mathcal{O}(\alpha^3)$$

Defining $h = \frac{Hv}{\|g\|}$ we can simplify

$$\text{PertAlign} = 1 - \frac{\alpha^2}{2} \left\| h - \frac{g^T h g}{\|g\|^2} \right\|^2$$

$$= 1 - \frac{\alpha^2}{2} \left\| h - (h^T \hat{g}) \hat{g} \right\|^2$$

where $\hat{g} = \frac{g}{\|g\|}$ is the unit gradient direction. Further, we can write

$$1 - \text{PertAlign} \approx \frac{\alpha^2}{2} \|h_{\perp g}\|^2$$

$\square$

*Remark* A.2. The conclusion of Lemma 4.1 also holds when the input is augmented with small random noise, i.e., $x' = x + \eta$, since such perturbations do not alter the logic of the proof.

**Corollary A.3** (Special case $v = g$). *If $v$ is aligned with the gradient, Lemma 4.1 reduces to*

$$\text{PertAlign} = 1 - \frac{\alpha^2}{2} \|H\hat{g} - k\hat{g}\|^2 + \mathcal{O}(\alpha^3), \tag{7}$$

*where $\hat{g} = g/\|g\|$ and $k = \frac{g^T H g}{\|g\|^2}$.*

*Remark* A.4. In this case, if the loss surface is locally linear in the gradient direction, equation 7 can be rewritten as

$$1 - \text{PertAlign} \approx \frac{\alpha^2}{2} \|(H - kI)g\|^2.$$

Thus the gradient satisfies $Hg \approx kg$, i.e., $\hat{g}$ is approximately an eigenvector of $H$, and PertAlign $\approx$ 1. If $\hat{g}$ is not aligned with any eigenvector, then $H\hat{g}$ points in a different direction from $\hat{g}$, implying the gradient direction changes after a small step.

**Lemma A.5** (Upper bound on misalignment). *For $v = g$,*

$$1 - \text{PertAlign} \leq \frac{\alpha^2}{2} \left( \|H\|^2 - k^2 \right) \leq \frac{\alpha^2}{2} \|H\|^2.$$

*Proof.* This follows directly from equation 6, using

$$\max_g \frac{\|Hg\|^2}{\|g\|^2} = \|H\|^2,$$

and the fact that $k^2 \geq 0$. $\square$

### A.4 STEP-SIZE SELECTION FROM ALIGNMENT APPROXIMATION

**Proposition A.6** (Optimal $\alpha$ for gradient ascent $v = \alpha g$). *Approximating the loss landscape locally as quadratic,*

$$\alpha^* = \begin{cases} \min \left( \alpha_{\max}, \dfrac{\alpha_0}{1 - \frac{\|g'\|}{\|g\|} \cdot \text{PertAlign}} \right), & \frac{\|g'\|}{\|g\|} \cdot \text{PertAlign} < 1, \\ \alpha_{\max}, & otherwise. \end{cases}$$

*where $g' = g + \alpha Hg$ is the perturbed gradient, $\alpha_{\max}$ is the maximum step size budget, and $\alpha_0$ is a fixed numerator.*

*Proof.* By approximating the loss landscape as a quadratic function, we can use alignment to find better candidates for step size ($\alpha$):

$$L(x + \delta) = L(x) + g^T \delta + \frac{1}{2} \delta^T H \delta$$

Using $\delta = \alpha g$, substitute in the equation above:

$$L(x + \alpha g) = L(x) + \alpha \|g\|^2 + \frac{\alpha^2}{2} g^T H g$$

We can find the optimal $\alpha$ as

$$\frac{\partial L(x + \alpha g)}{\partial \alpha} = 0 = \|g\|^2 + \alpha g^T H g \implies \alpha^* = -\frac{\|g\|^2}{g^T H g}$$

Using Taylor's approximation of the gradient:

$$\nabla_x L(x + \alpha g) = \nabla_x L(x) + \nabla_x^2 L(x)\delta = g' = g + \alpha H g$$

$$g^T g' = \|g\|^2 + \alpha g^T H g \implies g^T H g = \frac{g^T g' - \|g\|^2}{\alpha}$$

Substituting this into the step size equation:

$$\alpha^* = -\frac{\alpha \|g\|^2}{g^T g' - \|g\|^2} = \frac{\alpha}{1 - \frac{g^T g'}{\|g\|^2}}$$

Note that

$$\frac{g^T g'}{\|g\|^2} = \frac{\|g'\|}{\|g'\|} \frac{g^T g'}{\|g\|\|g\|} = \frac{\|g'\|}{\|g\|} \cdot \text{PertAlign}$$

Which gives us,

$$\alpha^* = \frac{\alpha}{1 - \frac{\|g'\|}{\|g\|} \cdot \text{PertAlign}}.$$

Note that, since our objective is to *maximize* the loss, the optimal step size must be strictly positive (i.e., $\alpha^* > 0$). If the $\alpha^*$ obtained from $\frac{dL}{d\alpha} = 0$ is negative, this implies $g^\top H g > 0$, corresponding to an *upward curvature* in the perturbation direction. In such cases, the computed $\alpha^*$ would lead toward a local minimum of the loss rather than a maximum. Therefore, to ensure loss maximization, we set the step size to the largest permissible value, namely, the maximum perturbation budget.

Because the quadratic approximation underlying this derivation becomes unreliable for large step sizes, we explicitly cap $\alpha^*$ by the perturbation budget. Moreover, to further stabilize the approximation in practice, we fix the numerator of the step-size formula to a constant value $\alpha_0$. This choice does not affect the validity of the proof as long as $\alpha_0$ remains sufficiently small.

Putting these considerations together, the final practical form of $\alpha^*$ is:

$$\alpha^* = \begin{cases} \min\left(\alpha_{\max}, \dfrac{\alpha_0}{1 - \dfrac{\|g'\|}{\|g\|} \cdot \text{PertAlign}}\right), & \dfrac{\|g'\|}{\|g\|} \cdot \text{PertAlign} < 1, \\ \alpha_{\max}, & \text{otherwise,} \end{cases}$$

where $\alpha_{\max}$ is the perturbation budget, $\alpha_0$ is the fixed numerator, $g'$ is the backpropagation gradient on the adversarial example, and PertAlign denotes the alignment metric defined in Equation 2.

$\square$

**Proposition A.7** (Optimal $\alpha$ for sign-gradient $v = \alpha p$). *For perturbation $v = \alpha p = \alpha \, \text{sign}(g)$,*

$$\alpha^* = \begin{cases} \min\left(\alpha_{\max}, \frac{\alpha_0}{1 - \frac{p^T g'}{\|g\|_1}}\right), & \frac{p^T g'}{\|g\|_1} < 1, \\ \alpha_{\max}, & \text{otherwise.} \end{cases}$$

*where $g' = g + \alpha H p$, $\|\cdot\|_1$ is the $\ell_1$ norm, $\alpha_{\max}$ is the maximum step size budget, and $\alpha_0$ is a fixed numerator..*

*Proof.* The loss after perturbation becomes:

$$L(x + \alpha p) = L(x) + \alpha g^T p + \frac{\alpha^2}{2} p^T H p.$$

Solving $\frac{dL}{d\alpha} = 0$ yields:

$$\alpha^* = -\frac{g^T p}{p^T H p}.$$

Using $g' = g + \alpha H p$, we have:

$$p^T g' = p^T g + \alpha p^T H p \quad \Longrightarrow \quad p^T H p = \frac{p^T g' - p^T g}{\alpha}.$$

Substituting yields:

$$\alpha^* = \frac{\alpha g^T p}{p^T g - p^T g'}.$$

Since $p^T g = \|g\|_1$:

$$\alpha^* = \frac{\alpha}{1 - \frac{p^T g'}{\|g\|_1}}.$$

Since our objective is to *maximize* the loss, the optimal step size must satisfy $\alpha^* > 0$. If the $\alpha^*$ obtained from the stationary condition $\frac{dL}{d\alpha} = 0$ is negative, this implies $p^\top H p > 0$, corresponding to an *upward curvature* along the perturbation direction. In such cases, the computed $\alpha^*$ would lead toward a local minimum of the loss rather than its maximum. To avoid this, we set the step size to the largest permissible value, namely the perturbation budget.

Because the quadratic approximation underlying this derivation becomes unreliable for large step sizes, we explicitly cap $\alpha^*$ by the perturbation budget. Moreover, to further stabilize the approximation in practice, we fix the numerator of the step-size formula to a constant value $\alpha_0$. This choice does not affect the validity of the proof as long as $\alpha_0$ remains sufficiently small.

Putting these considerations together, the final practical form of $\alpha^*$ is:

$$\alpha^* = \begin{cases} \min\left(\alpha_{\max}, \dfrac{\alpha_0}{1 - \frac{p^\top g'}{\|g\|_1}}\right), & \dfrac{p^\top g'}{\|g\|_1} < 1, \\ \\ \alpha_{\max}, & \text{otherwise,} \end{cases}$$

where $\alpha_{\max}$ denotes the perturbation budget, $\alpha_0$ is the fixed numerator, $g'$ is the backpropagation gradient computed on the adversarial example, and $p$ is the perturbation direction.

$\square$

## A.5 ALTERNATIVE PERTALIGN VARIANT (ABLATION)

**Lemma A.8** (First-order Hessian correlation for sign alignment)**.** *Let $p = \text{sign}(g)$. Define*

$$\text{AltPertAlign} = \beta \cos(p, g'), \quad g' = g + \alpha H p.$$

*For $\alpha \ll 1$,*

$$1 - \text{AltPertAlign} \approx \alpha \left( \frac{p^T H p}{\|g\|_1} - \frac{g^T H p}{\|g\|^2} \right),$$

*with $\beta = \frac{\sqrt{d}\|g\|}{\|g\|_1}$.*

*Proof.* Let $L(x) := \mathcal{L}(f_\theta(x), y)$. Using a second-order Taylor expansion,

$$\nabla_x L(x + \delta) = \nabla_x L(x) + H\delta + \mathcal{O}(\|\delta\|^2) = g + \alpha H p + R, \quad R = \mathcal{O}(\alpha^2).$$

Let $g' := \nabla_x L(x + \delta)$. Then,

$$p^T g' = p^T g + \alpha p^T H p + p^T R, \tag{8}$$

$$\|g'\|^2 = \|g\|_2^2 + 2\alpha g^T H p + \alpha^2 \|Hp\|^2 + 2g^T R + \mathcal{O}(\alpha^3). \tag{9}$$

Using equations 8 and 9,

$$\text{AltPertAlign} = \beta \frac{p^T g + \alpha p^T H p + p^T R}{\|p\| \sqrt{\|g\|^2 + 2\alpha g^T H p + \alpha^2 \|Hp\|^2 + 2g^T R + \mathcal{O}(\alpha^3)}}$$

$$= \beta \frac{\|g\|_1}{\|p\| \|g\|} \frac{1 + \alpha \frac{p^T H p}{\|g\|_1} + \frac{p^T R}{\|g\|_1}}{\sqrt{1 + 2\alpha \frac{g^T H p}{\|g\|^2} + \alpha^2 \frac{\|Hp\|^2}{\|g\|^2} + 2\frac{g^T R}{\|g\|^2} + \mathcal{O}(\alpha^3)}}$$

Using the expansion $\frac{1}{\sqrt{1+\epsilon}} = 1 - \frac{\epsilon}{2} + \mathcal{O}(\epsilon^2)$ and setting $\epsilon = 2\alpha \frac{g^T H p}{\|g\|^2} + \alpha^2 \frac{\|Hp\|^2}{\|g\|^2} + 2\frac{g^T R}{\|g\|^2} + \mathcal{O}(\alpha^3)$,

$$\text{AltPertAlign} \approx \beta \frac{\|g\|_1}{\|p\| \|g\|} \left(1 + \alpha \frac{p^T H p}{\|g\|_1} + \frac{p^T R}{\|g\|_1}\right) \left(1 - \alpha \frac{g^T H p}{\|g\|^2} - \frac{\alpha^2}{2} \frac{\|Hp\|^2}{\|g\|^2} - \frac{g^T R}{\|g\|^2}\right)$$

Neglecting second-order terms,

$$\text{AltPertAlign} \approx \beta \frac{\|g\|_1}{\|p\| \|g\|} \left(1 + \alpha \left(\frac{p^T H p}{\|g\|_1} - \frac{g^T H p}{\|g\|^2}\right)\right)$$

Choosing $\beta = \frac{\|p\| \|g\|}{\|g\|_1}$, we obtain

$$1 - \text{AltPertAlign} \approx \alpha \left(\frac{p^T H p}{\|g\|_1} - \frac{g^T H p}{\|g\|^2}\right)$$

Further, since $p \in \{-1, 0, 1\}^d$,

$$\beta = \frac{\sqrt{d} \|g\|_2}{\|g\|_1} \tag{10}$$

$\square$

# B  BASELINES

We provide details about the hyperparameters of the baseline models here.

## B.1  FGSM

The Fast Gradient Sign Method (FGSM) introduced by Goodfellow et al. (2015) can be used directly in Equation 1. FGSM consists of a single-step update using the sign of the gradient:

$$x_{\text{adv}} = x + \epsilon \cdot \text{sign} \left( \nabla_x \mathcal{L}(f_\theta(x), y) \right)$$

Madry et al. (2019) and Wong et al. (2020) have observed that FGSM adversarial training suffers from Catastrophic Overfitting. We include FGSM adversarial training despite this limitation as baseline for other models ability to overcome CO.

## B.2  FGSM-RS

Wong et al. (2020) introduced Fast Adversarial Training, in which each attack starts by adding uniform noise $\eta \sim \mathcal{U}(-\epsilon, +\epsilon)^d$ to the clean input. This perturbed sample then serves as the starting point for a standard FGSM update. The resulting perturbation $\delta$ is constrained within the $\ell_\infty$-norm ball of radius $\epsilon = 8/255$. The method also requires a step-size hyperparameter, for which the authors recommend $\alpha = 10/255$.

## B.3  GRADALIGN

Andriushchenko & Flammarion (2020) proposed a regularization method that maximizes gradient alignment within the perturbation set. They define the following local linearity metric of the loss function:

$$\text{GradAlign}(x, y, \theta) := \cos \left( \nabla_x \mathcal{L}\big(f_\theta(x), y\big), \nabla_x \mathcal{L}\big(f_\theta(x + \eta), y\big) \right), \tag{11}$$

where $\eta \sim \mathcal{U}(-\epsilon, +\epsilon)^d$. They introduce a regularizer $\Omega(x, y, \theta) = 1 - \text{GradAlign}(x, y, \theta)$ and optimize the objective $\mathcal{L} + \lambda \Omega$, with $\lambda = 0.2$ fixed across all architectures and datasets, following the authors' recommendations for CIFAR-10 and CIFAR-100. Here, $\mathcal{L}$ denotes the cross-entropy loss computed on adversarial examples generated by FGSM-RS with $\alpha = 1.25 \times \epsilon$.

## B.4  ZEROGRAD & MULTIGRAD

Golgooni et al. (2021) proposed ZeroGrad, a method that sets a threshold $q$ and zeros out the components of the input gradient whose absolute value falls below this threshold, thereby producing a more robust single-step perturbation. The method uses a random initialization $\eta \sim \mathcal{U}(-\epsilon, +\epsilon)^d$ and a step size $\alpha$.

ZeroGrad is highly sensitive to the choice of $q$ across datasets, architectures, and training settings. For a fair comparison of generalizability among baselines, we follow the authors' recommendations and set $\alpha = 2 \times \epsilon$ and $q = 0.35$ for CIFAR-10, and $q = 0.45$ for CIFAR-100 and all other datasets.

Golgooni et al. (2021) also introduced MultiGrad, in which an identical batch is concatenated $N$ times, each copy initialized with $\eta \sim \mathcal{U}(-\epsilon, +\epsilon)^d$. Perturbations are then retained only in directions where all samples agree on the gradient sign. MultiGrad is less sensitive to hyperparameter choices, so we set $N = 3$ and $\alpha = 2 \times \epsilon$ in all experiments as per recommended by authors.

In Appendix C.2 we analyze these methods through the lens of second order optimization, providing a fresh new look explaining how ZeroGrad and MultiGrad mitigate CO.

## B.5  N-FGSM

de Jorge et al. (2022) proposed an FGSM variant that initializes with larger random noise and omits clipping of the final perturbation $\delta$ to the $\ell_\infty$-norm ball of radius $\epsilon$. Following the authors' recommendation, we set $k = 2 \times \epsilon$ for the initialization $\eta \sim \mathcal{U}(-k, +k)^d$ and use a step size of $\alpha = \epsilon$.

### B.6  ATAS

Huang et al. (2022) introduced Adversarial Training with Adaptive Step Sizes (ATAS), which maintains a moving average of the squared $\ell_2$-norm of the gradient:

$$v_i^j = \beta v_i^{j-1} + (1 - \beta) \left\| \nabla_{\tilde{x}_i} \mathcal{L}\big(f_\theta(\tilde{x}_i), y_i\big) \right\|_2^2,$$

where $\tilde{x}_i$ is the initialization of $x_i$, and $\beta$ is a momentum factor that stabilizes the step size.

The per-example step size $\alpha_i^j$ at epoch $j$ is then adjusted inversely to $v_i^j$:

$$\alpha_i^j = \frac{\gamma}{c + \sqrt{v_i^j}},$$

where $\gamma$ is a predefined learning rate and $c$ is a constant that prevents $\alpha_i^j$ from becoming excessively large. Following the authors' recommendations for CIFAR-10 and CIFAR-100, we set $\beta = 0.5$, $c = 0.01$, and $\gamma = 2c\epsilon$.

### B.7  AAER

Lin et al. (2024) identified *Abnormal Adversarial Examples* (AAEs) as a primary cause of CO. Unlike *Normal Adversarial Examples* (NAEs), AAEs exhibit lower loss than their corresponding clean samples:

$$x^{\text{AAE}} := \mathcal{L}\big(f_\theta(x + \eta), y\big) > \mathcal{L}\big(f_\theta(x + \eta + \delta), y\big),$$
$$x^{\text{NAE}} := \mathcal{L}\big(f_\theta(x + \eta), y\big) \leq \mathcal{L}\big(f_\theta(x + \eta + \delta), y\big).$$

To mitigate the adverse effect of AAEs, they introduce a regularizer. For a batch of size $m$, let $n$ be the number of AAEs. The following terms penalize anomalous variation in AAEs and disparities in logits:

$$\text{AAE-CE} = \frac{1}{n} \sum_{i=1}^{n} \left[ \mathcal{L}\left( f_\theta(x_i^{\text{AAE}} + \eta), y_i \right) - \mathcal{L}\left( f_\theta(x_i^{\text{AAE}} + \eta + \delta), y_i \right) \right],$$

$$\text{AAE-L2} = \frac{1}{n} \sum_{i=1}^{n} \left\| f_\theta(x_i^{\text{AAE}} + \eta + \delta) - f_\theta(x_i^{\text{AAE}} + \eta) \right\|_2^2,$$

$$\text{NAE-L2} = \frac{1}{m - n} \sum_{j=1}^{m-n} \left\| f_\theta(x_j^{\text{NAE}} + \eta + \delta) - f_\theta(x_j^{\text{NAE}} + \eta) \right\|_2^2.$$

The *Abnormal Adversarial Examples Regularization* (AAER) term is defined as:

$$\text{AAER} = \left( \lambda_1 \cdot \frac{n}{m} \right) \cdot \left( \lambda_2 \cdot \text{AAE-CE} + \lambda_3 \cdot \max\left( \text{AAE-L2} - \text{NAE-L2}, 0 \right) \right).$$

Two variants were proposed: RS-AAER, based on FGSM-RS (Wong et al., 2020), and N-AAER, based on N-FGSM (de Jorge et al., 2022), both augmented with the AAER regularization term. In our experiments, we adopt N-AAER as the best-performing variant. Following the authors' recommendations for a fair generalizability comparison, we fix $\lambda_1 = 1$, $\lambda_2 = 1.5$, and $\lambda_3 = 0.15$ across all datasets and architectures.

### B.8  ELLE

Rocamora et al. (2024) proposed *Efficient Local Linearity Enforcement* (ELLE), a regularization term designed to encourage local linearity. ELLE mitigates CO not only in standard AT evaluations, but also in more challenging scenarios such as large adversarial perturbations and extended training schedules. The regularization term is theoretically linked to the curvature of the loss function and requires three forward passes to compute, as follows:

$$x_a, \, x_b \sim x + \mathcal{U}\big( -\epsilon, +\epsilon \big)^d,$$

$$\alpha \sim \mathcal{U}(0, 1),$$

where $d$ is the input dimensionality. A convex combination of $x_a$ and $x_b$ is then formed:

$$x_c = (1 - \alpha) \cdot x_a + \alpha \cdot x_b.$$

The ELLE penalty is given by:

$$E_{\text{lin}} = \big| \mathcal{L}\big(f_\theta(x_c), y^i\big) - (1 - \alpha) \cdot \mathcal{L}\big(f_\theta(x_a), y^i\big) - \alpha \cdot \mathcal{L}\big(f_\theta(x_b), y^i\big) \big|^2.$$

Because the ELLE term can take on large values, an excessively high coefficient may cause numerical overflow in the model weights. Following the authors' recommendations for CIFAR-10 and CIFAR-100, we set the regularization coefficient to $\lambda = 1000$ in all experiments.

### B.9   PGD

Madry et al. (2019) proposed Projected Gradient Descent (PGD) as a multi-step adversarial attack, widely regarded as a strong first-order adversary for both evaluation and training. PGD iteratively applies the gradient-sign method to maximize the loss with respect to the input, projecting intermediate updates back onto the perturbation set to ensure the adversarial example remains within the allowed $\ell_p$-norm constraint.

Given a clean example $x$, PGD generates an adversarial example by initializing from a random perturbation within $\Delta$ and performing $K$ steps:

$$x^{t+1} = \Pi_{x+\Delta} \left( x^t + \alpha \cdot \text{sign} \left( \nabla_x \mathcal{L}(f_\theta(x^t), y) \right) \right),$$

where $\Pi_{x+\Delta}(\cdot)$ denotes the projection onto $x + \Delta$, and $\alpha$ is the step size.

In our experiments, we adopt PGD-10 for generating adversarial examples during training with random starts.

### B.10   TRADES

Zhang et al. (2019) proposed TRadeoff-inspired Adversarial DEfense via Surrogate-loss minimization (TRADES), a regularized adversarial training framework that explicitly balances clean accuracy and robust accuracy. The method augments the standard adversarial training objective with a penalty that minimizes the Kullback–Leibler (KL) divergence between the model's output distributions on clean and adversarial examples, thereby encouraging prediction consistency under perturbations.

Formally, TRADES solves:

$$\min_\theta \mathbb{E}_{(x,y)\sim\mathcal{D}} \left[ \mathcal{L}\big(f_\theta(x), y\big) + \frac{1}{\lambda} \max_{\delta \in \Delta} D_{\text{KL}} \left( f_\theta(x) \,\|\, f_\theta(x + \delta) \right) \right], \tag{12}$$

where $\mathcal{L}$ denotes the standard cross-entropy loss on clean inputs, $\Delta$ is typically an $\ell_\infty$-norm ball of radius $\epsilon$, and $\lambda$ is a trade-off hyperparameter controlling the balance between natural and robust accuracy.

In practice, the inner maximization is performed using PGD-10, but with the KL divergence term replacing the cross-entropy loss used in standard PGD-based adversarial training. Following the authors' recommendations for CIFAR-10, we set $\beta = 1/\lambda = 6$ in all experiments.

## C  REINTERPRETING PREVIOUS WORK VIA OUR FRAMEWORK

Here we dive deeper into previous works and and focus on some closely related ideas. In Appendix C.1 we explain the key differences between GradAlign and our proposed PertAlign metric. In Appendix C.2 we provide a **new perspective** on ZeroGrad and MultiGrad, and elaborate how they manage to mitigate CO. Appendix C.3 examines the relation between SORA and multi-step adversarial training in a simplified setting.

### C.1  GRADALIGN VERSUS PERTALIGN

Andriushchenko & Flammarion (2020) introduced the GradAlign regularizer, which measures the cosine similarity between the gradient of the loss with respect to the clean input and the gradient with respect to the clean input perturbed by random noise:

$$\texttt{GradAlign} := \cos\left(\nabla_x \mathcal{L}(f_\theta(x), y), \nabla_x \mathcal{L}(f_\theta(x + \eta), y)\right), \qquad \eta \sim \mathcal{U}(-\epsilon, +\epsilon)^d.$$

Lemma 4.1, which forms the theoretical basis for PertAlign, can also be applied to GradAlign. Without loss of generality, we can omit the explicit $\eta$ in Equation 2, viewing the noise $\eta$ in GradAlign as the arbitrary perturbation $v$ in Equation 2. This allows us to interpret GradAlign through a second-order approximation lens. We believe this perspective explains why PertAlign and GradAlign exhibit similar trends in Figure 3.

However, despite the similarity of their underlying metrics, our method differs from GradAlign in three key aspects:

1. **Motivation.** The primary motivation behind GradAlign is to increase gradient alignment *within* the perturbation budget. In contrast, our focus is on promoting the *linearity* of the loss surface along gradient directions. This imposes fewer constraints on the model and facilitates robustness in critical directions while better preserving clean accuracy.

2. **Usage.** GradAlign is designed as a *training regularizer*. In our case, PertAlign is used solely as a monitoring metric, enabling low-cost (practically zero) per-batch prediction of the CO status without influencing the training dynamics directly.

3. **Computation.** Since GradAlign serves as a regularizer, it must be computed before backpropagation to update the weights. In contrast, PertAlign leverages gradients already computed for attack generation and backpropagation (extended one layer to include the inputs), adding virtually no extra computational or memory overhead (Figure 6).

### C.2  ZEROGRAD & MULTIGRAD FROM A SECOND ORDER PERSPECTIVE

Golgooni et al. (2021) proposed ZeroGrad as an adversarial training method to prevent Catastrophic Overfitting (CO). It works by zeroing the perturbation components in directions where the corresponding gradient elements have very small magnitude. In this section, we examine ZeroGrad from a second-order theoretical perspective.

The ZeroGrad perturbation can be modeled as:

$$v = \alpha \odot p, \quad \alpha = \begin{pmatrix} \alpha_1 \\ \vdots \\ \alpha_d \end{pmatrix}, \quad p = \begin{pmatrix} \text{sign}(g_1) \\ \vdots \\ \text{sign}(g_d) \end{pmatrix},$$

where $\odot$ denotes the element-wise product. Each component $\alpha_i$ is determined by:

$$\forall i \in \{1, \cdots, d\}: \quad \alpha_i = \begin{cases} 1 & \text{if } \frac{|g_i|}{\|g\|} > \tau, \\ 0 & \text{otherwise.} \end{cases}$$

Since $\|v\| \ll \|x\|$, we can approximate the loss at the adversarial example $x + v$ using a second-order Taylor expansion:

$$L(x + v) = L(x + \alpha \odot p) \approx L(x) + g^\top (\alpha \odot p) + \frac{1}{2} (\alpha \odot p)^\top H (\alpha \odot p),$$

where $g = \nabla_x L(x)$ and $H = \nabla_x^2 L(x)$.

For the attack to succeed, we require:

$$L(x + \alpha \odot p) > L(x),$$

which is equivalent to:

$$g^\top(\alpha \odot p) + \frac{1}{2}(\alpha \odot p)^\top H(\alpha \odot p) > 0.$$

Since $H$ is symmetric, its eigendecomposition is:

$$H = Q^\top \Lambda Q,$$

where $Q$ is orthogonal and $\Lambda$ is diagonal with eigenvalues $\lambda_i$. Substituting this into the inequality yields:

$$g^\top(\alpha \odot p) + \frac{1}{2}\left(Q(\alpha \odot p)\right)^\top \Lambda \, Q(\alpha \odot p) = g^\top(\alpha \odot p) + \frac{1}{2}h^\top \Lambda h,$$

where $h = Q(\alpha \odot p)$. Component-wise, this becomes:

$$\sum_{i=1}^{d} \alpha_i p_i g_i + \frac{1}{2}\lambda_i h_i^2 = \sum_{i=1}^{d} \alpha_i |g_i| + \frac{1}{2}\lambda_i h_i^2.$$

For each term $i$ we have:

$$\alpha_i |g_i| + \frac{1}{2}\lambda_i h_i^2. \tag{13}$$

Because $\alpha_i |g_i| \geq 0$ and $h_i^2 \geq 0$, the sign of equation 13 depends on the eigenvalue $\lambda_i$. If $\lambda_i < 0$ and

$$\lambda_i < -\frac{2\alpha_i |g_i|}{h_i^2},$$

then the $i$-th term is negative. If enough such terms are negative, the total sum becomes negative, meaning the adversarial example has lower loss than the clean sample—an *Abnormal Adversarial Example* (AAE). As noted by Lin et al. (2024), AAEs are closely linked to the onset of CO.

For components where $|g_i|/\|g\| \ll 1$, this condition is more easily satisfied, increasing the likelihood of generating AAEs. ZeroGrad mitigates this by setting $\alpha_i = 0$ whenever $|g_i|/\|g\| < \tau$, thereby avoiding perturbations in directions that could reduce the loss.

However, the magnitudes of $\lambda_i$ and $g_i$ vary substantially across datasets, models, and training stages. This variability directly affects the chosen $\tau$ hyperparameter (which closely relates to $q_{\text{val}}$ in Golgooni et al. (2021)), which can lead to limited generalizability in diverse scenarios.

Golgooni et al. (2021) also proposed MultiGrad which tries to overcome shortcomings of ZeroGrad by only perturbing in directions where different random starts all show same gradient sign, but this is at the cost of more computational cost and memory consumption.

### C.3 PGD-2 AND SORA

PertAlign computes the cosine similarity between gradients from the first two iterations of a PGD attack. Significant divergence between these gradients indicates a highly distorted and non-linear loss surface. High alignment between consecutive PGD gradients suggests that the resulting perturbation could be approximated by a single-step attack. Conversely, when the loss surface becomes distorted, the alignment between PGD iterations decreases, and the optimization path can no longer be inferred from the initial gradient direction.

PGD variants, including PGD-2, avoid catastrophic failure by refining their results over multiple iterations. As shown in Figure 11, during catastrophic overfitting (CO), PertAlign collapses to zero, indicating that the second attack iteration moves in a direction nearly orthogonal to the first. This explains the substantial accuracy gap between single-step and multi-step methods after CO.

However, the small fixed step size used in PGD, while beneficial in some contexts, introduces limitations. Fixed step sizes can significantly reduce convergence speed compared to adaptive approaches.

Auto-PGD (APGD) (Croce & Hein, 2020) addresses this by adapting step sizes based on optimization progress and restarting from the best-found point when step sizes are reduced. SORA builds upon these insights by using curvature information to estimate optimal step sizes adaptively.

Another limitation of standard PGD is its use of a fixed $\epsilon$ budget. As noted by Ding et al. (2020), adaptive selection of $\epsilon$ for individual data points, treating it as a margin, can improve performance. SORA avoids the pitfalls of fixed-$\epsilon$ training identified by Ding et al. (2020), including the potential reduction of margins between clean samples and decision boundaries.

### C.4 FGSM-CKPT

Kim et al. (2020) observed that, upon the onset of CO, the loss landscape becomes distorted in the direction of single-step perturbations. They proposed FGSM-CKPT, which partitions the attack step size $\alpha$ into $c$ discrete fractions. For each scaled step size $\frac{i\alpha}{c}$, the perturbed input $x + \frac{i\alpha}{c}$ is fed to the model, and the index $i$ yielding the lowest accuracy is selected; the model is then updated using that perturbation.

Although effective in some cases, FGSM-CKPT requires multiple forward passes per training step, resulting in high computational cost. Moreover, restricting the step size to a discrete set limits flexibility in selecting the optimal value. In this work, we analyze the loss landscape in greater detail and, leveraging observed distortion patterns, propose an adaptive mechanism that selects a more accurate step size with effectively zero additional cost.

# D  DATASETS

In this section we explore the efficacy of different methods on a number of different datasets. We provide a brief overview of the key aspects of each dataset in Table 2. The distribution of labels in each dataset can be seen in Figure 7.

Table 2: Datasets overview.

| Dataset | Dimensions | # Classes | # Training Samples | # Test Samples |
|---|---|---|---|---|
| CIFAR-10 | $3 \times 32 \times 32$ | 10 | 50,000 | 10,000 |
| CIFAR-100 | $3 \times 32 \times 32$ | 100 | 50,000 | 10,000 |
| TINYIMAGENET | $3 \times 64 \times 64$ | 200 | 100,000 | 10,000 |
| PATHMNIST | $3 \times 28 \times 28$ | 9 | 89,996 | 7,180 |
| TISSUEMNIST | $1 \times 28 \times 28$ | 8 | 165,466 | 47,280 |

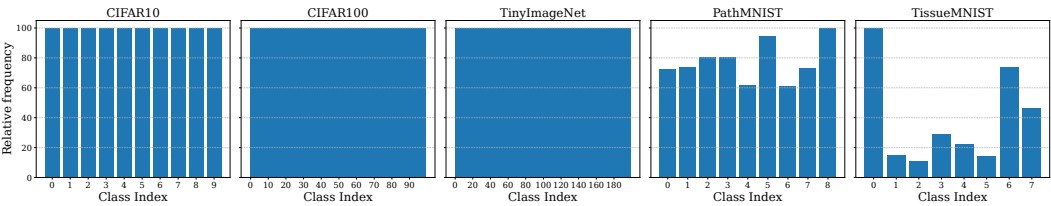

Figure 7: Class distributions across datasets. CIFAR-10 and CIFAR-100 exhibit balanced class distributions, whereas PATHMNIST and TISSUEMNIST are imbalanced, with TISSUEMNIST showing the most pronounced imbalance.

Szegedy et al. (2014) define adversarial examples as *imperceptible* non-random perturbations added to test images, capable of arbitrarily changing the network's prediction. In general, imperceptibly tiny perturbations of a given image do not normally change the underlying class for most computer vision problems (Szegedy et al., 2014). In order to abide by this convention of *imperceptibly*, accompanied by the desire to not accidentally change the true label of images by distorting them too much, we reduce the value of $\epsilon$ to $3/255$ for grayscale images of the TISSUEMNIST dataset. For the remaining datasets we use an $\epsilon$ value of $8/255$ similar to all previous works. Samples from each dataset alongside their corresponding adversarial perturbations can be viewed in Figure 8.

## D.1  CIFAR-10

The CIFAR-10 dataset (Krizhevsky, 2009) consists of 60,000 $32 \times 32$ color images across 10 classes, with 6,000 images per class. The dataset is divided into 50,000 training images and 10,000 test images. The test set contains exactly 1,000 randomly-selected images from each class, while the training batches contain the remaining images with some variation in class distribution per batch, though each class has exactly 5,000 training images in total.

Following the instructions of Wong et al. (2020), we use padding of size 4 and perform random crops, in addition to random horizontal flips on the training data. Both the training and test data are normalized with respect to the mean and standard deviation of the training data. For all models and architectures we use an SGD optimizer with momentum 0.9 and weight decay $5 \times 10^{-4}$. A simple cyclic learning rate schedules the learning rate linearly from 0.01, to a maximum learning rate of 0.2, and back down to 0.01.

## D.2  CIFAR-100

The CIFAR-100 dataset (Krizhevsky, 2009) contains 60,000 $32 \times 32$ color images across 100 fine-grained classes, which are grouped into 20 superclasses. Each class contains 600 images, with 500

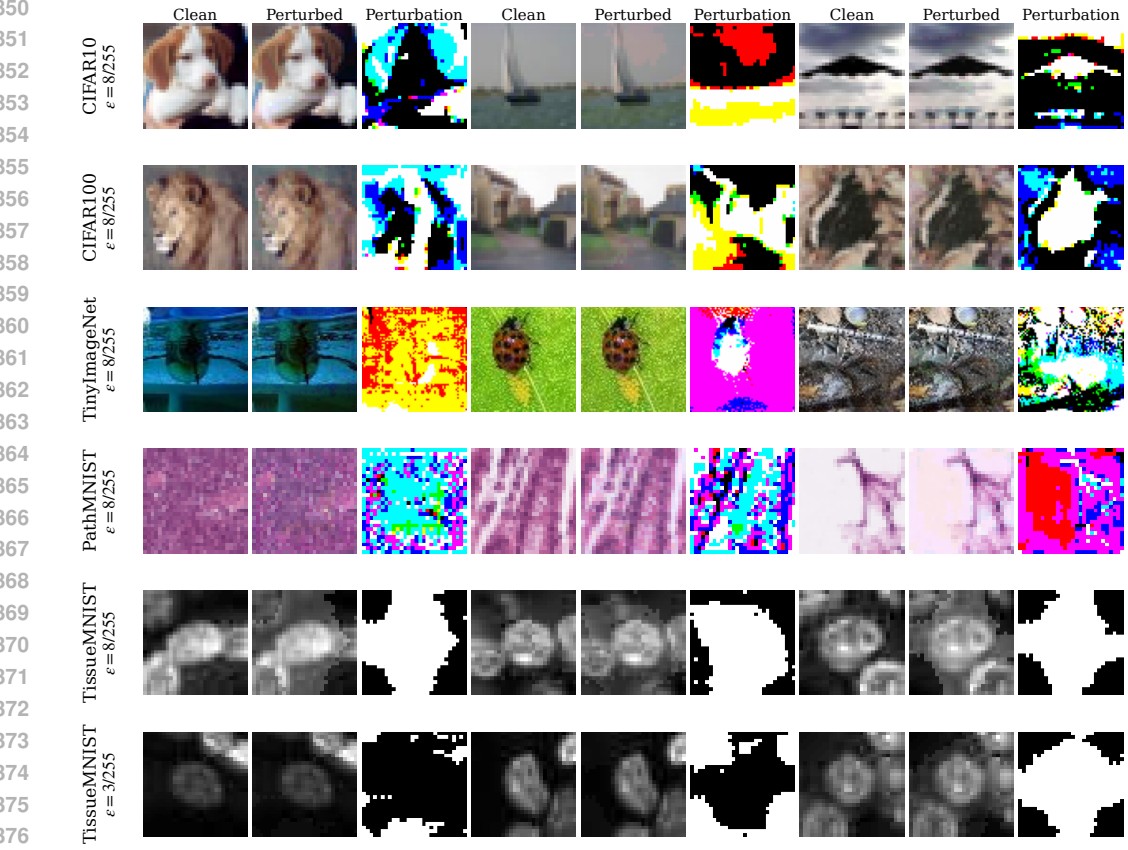

Figure 8: Dataset samples and their corresponding FGSM adversarial examples for a model adversarially trained with the SORA method. Each row corresponds to a dataset (CIFAR-10, CIFAR-100, TINYIMAGENET, PATHMNIST, and TISSUEMNIST, respectively). The "Clean" columns show the original images, the "Perturbed" columns show the FGSM adversarial examples, and the "Perturbation" columns visualize the added perturbations, amplified for clarity.

training and 100 testing images per class. Each image is annotated with both a fine label (specific class) and a coarse label (superclass).

Following the instructions of Wong et al. (2020), we use padding of size 4 and perform random crops, in addition to random horizontal flips on the training data. Both the training and test data are normalized with respect to the mean and standard deviation of the training data. For all models and architectures we use an SGD optimizer with momentum $0.9$ and weight decay $5 \times 10^{-4}$. A simple cyclic learning rate schedules the learning rate linearly from $0.01$, to a maximum learning rate of $0.2$, and back down to $0.01$.

### D.3 TINYIMAGENET

The TINYIMAGENET dataset (Le & Yang, 2015) is a downsampled version of IMAGENET (Deng et al., 2009) containing 100,000 $64 \times 64$ color images across 200 classes. Each class has 500 training images, 50 validation images, and 50 test images, providing a more computationally manageable alternative to the full ImageNet dataset while maintaining the multi-class classification challenge.

Following the instructions of Wong et al. (2020) and Lin et al. (2024), we use padding of size 8 and perform random crops, in addition to random horizontal flips on the training data. Both the training and test data are normalized with respect to the mean and standard deviation of the training data. For all models and architectures we use an SGD optimizer with momentum $0.9$ and weight decay

$5 \times 10^{-4}$. A simple cyclic learning rate schedules the learning rate linearly from $0.01$, to a maximum learning rate of $0.2$, and back down to $0.01$.

### D.4 PATHMNIST

The PATHMNIST dataset (Yang et al., 2023) is a medical image classification benchmark for colorectal cancer detection. It contains 107,180 $28 \times 28$ color histological images across 9 tissue classes. The dataset is split into 89,996 training images, 10,004 validation images, and 7,180 test images. Unlike the balanced natural image datasets, PATHMNIST exhibits natural class imbalance reflective of real-world medical data distributions.

We use padding of size 2 and perform random rotations with 10 degrees, in addition to random horizontal flips on the training data. Both the training and test data are normalized with respect to the mean and standard deviation of the training data. For all models and architectures we use an SGD optimizer with momentum $0.9$ and weight decay $5 \times 10^{-4}$. To further stabilize training, we use a cosine annealing learning rate scheduler that starts from the initial learning rate of $0.05$ and the minimum learning rate of $10^{-3}$. This scheduler improves the results for all results.

### D.5 TISSUEMNIST

The TISSUEMNIST dataset (Yang et al., 2023) is based on the Brodely annotated tissue atlas of human gene expression, containing 236,386 $28 \times 28$ grayscale images of human kidney cortex cells across 8 tissue classes. The dataset is split into 165,466 training images, 23,640 validation images, and 47,280 test images. TissueMNIST exhibits significant class imbalance, making it particularly challenging for evaluation and representative of real-world medical imaging scenarios.

We use padding of size 2 and perform random rotations with 10 degrees, in addition to random horizontal flips on the training data. Both the training and test data are normalized with respect to the mean and standard deviation of the training data. For all models and architectures we use an SGD optimizer with momentum $0.9$ and weight decay $5 \times 10^{-4}$. To further stabilize training, we use a cosine annealing learning rate scheduler that starts from the initial learning rate of $0.05$ and the minimum learning rate of $10^{-3}$. This scheduler improves the results for all results.

# E ARCHITECTURES

In this section, we describe the architectures employed in our experiments. We include classical convolutional backbones such as ResNets (He et al., 2015), PreActResNets (He et al., 2016), and WideResNets (Zagoruyko & Komodakis, 2017), as well as more recent variants like SENets (Hu et al., 2019).

## E.1 RESNET

Residual Networks (ResNets) introduced by He et al. (2015) remain a cornerstone of deep learning due to their ability to train very deep models effectively. The key innovation is the residual connection, which mitigates the vanishing gradient problem by providing identity shortcuts for direct gradient propagation. This enables deeper networks to converge more reliably, enhancing representational capacity without significant optimization challenges. In our experiments, we use ResNet-18, which offers a balance between computational efficiency and representational power.

## E.2 PREACTRESNET

Pre-activation Residual Networks (PreActResNets) (He et al., 2016) refine the original ResNet architecture by placing batch normalization and ReLU activations before the convolutional layers. This modification improves gradient flow and facilitates optimization, particularly in deeper architectures. We evaluate PreActResNet-18, which maintains the overall structure of standard ResNets while benefiting from more stable training dynamics.

## E.3 WIDERESNET

Wide Residual Networks (WideResNets) (Zagoruyko & Komodakis, 2017) challenge the trend of increasing depth by instead widening residual blocks through additional channels. This approach enhances model capacity and generalization while reducing training time compared to deeper, narrower networks. We employ WideResNet-28-10, where 28 indicates the number of convolutional layers and 10 is the widening factor applied to channels in each residual block relative to a standard ResNet.

## E.4 SENET

Squeeze-and-Excitation Networks (SENets) (Hu et al., 2019) enhance convolutional architectures with channel-wise attention mechanisms. By adaptively recalibrating feature responses, SENets emphasize informative channels and suppress less useful ones, improving performance across various recognition tasks. We use SENet-18, which integrates squeeze-and-excitation blocks into a standard ResNet-18, providing an effective attention-enhanced backbone with minimal computational overhead.

# F ABLATIONS

## F.1 ABLATION ON SORA

We perform an ablation study to evaluate the contribution of each component of SORA to its overall performance. Experiments are conducted using PreActResNet-18 on the PATHMNIST dataset, which is more susceptible to CO than CIFAR-10 and CIFAR-100, and thus better exposes the effect of each component. As shown in Table 3, restricting the variability of the step-size magnitude significantly reduces robust accuracy, most notably, removing the optimal step-size selection yields the lowest PGD-10 accuracy.

Table 3: Ablation study on the components of SORA.

| Configuration | Clean | FGSM | PGD-10 |
|---|---|---|---|
| **SORA (baseline)** | 85.62 | 54.13 | **40.51** |
| – Without Random Sampling | **87.25** | 34.20 | 17.92 |
| – Clamping Step Size | 78.87 | 49.33 | 16.89 |
| – Without Optimal Step Size | 32.42 | **84.42** | 2.36 |

These results highlight the importance of both attack strength and directional variability, as discussed in Section 3, and demonstrate the necessity of adaptive step-size selection for challenging datasets.

## F.2 BATCH SIZE

We run our SORA algorithm with different batch sizes on the CIFAR-10 dataset. You can see the results for our method and NFGSM in Table 4.

Table 4: **Ablation on batch size.** The values in each cell correspond to clean, FGSM and PGD-10 accuracies.

| Method | Batch Size 32 | Batch Size 64 | Batch Size 128 | Batch Size 256 | Batch Size 512 |
|---|---|---|---|---|---|
| NFGSM | $73.40 \pm 0.08$ | $77.15 \pm 0.29$ | $79.05 \pm 0.21$ | $79.95 \pm 0.15$ | $78.69 \pm 0.21$ |
| | $46.02 \pm 0.16$ | $48.15 \pm 0.42$ | $48.73 \pm 0.44$ | $47.84 \pm 0.37$ | $46.91 \pm 0.09$ |
| SORA (Ours) | $74.00 \pm 0.40$ | $78.06 \pm 0.38$ | $79.95 \pm 0.01$ | $80.55 \pm 0.29$ | $79.51 \pm 0.24$ |
| | $46.25 \pm 0.36$ | $48.48 \pm 0.41$ | $48.72 \pm 0.30$ | $47.81 \pm 0.37$ | $47.09 \pm 0.36$ |

## F.3 DIFFERENT EPSILONS AT TRAINING

We trained our SORA algorithm with different values of $\epsilon$ during training. The results for this experiment are reported in Table 5. Since SORA uses adaptive step sizes, for smaller perturbation budgets it can fully exploit the allotted space. For larger values of epsilon it manages to detect the most effective step size to avoid CO and improve robustness.

## F.4 DIFFERENT EPSILONS AT TEST TIME

We report the accuracy of different methods against different values of $\epsilon$ for FGSM and PGD attacks in Table 6. Since FGSM training suffers from CO it dominates the other methdos in terms of FGSM accuracy but is not actually robust as indicated by its PGD accuracy. Other methods also struggle to match the performance of SORA across all values of $\epsilon$. Methods like ELLE and GradAlign perform better than our SORA for smaller $\epsilon$ values but their performance deteriorates rapidly when we move to larger values. SORA retains its accuarcy and adapts better to attacks with different perturbation budgets.

Table 5: **Ablation on training with different epsilons.** The values in each cell correspond to clean, FGSM and PGD-10 accuracies.

| Method | $\epsilon = {}^{4}/255$ | $\epsilon = {}^{8}/255$ | $\epsilon = {}^{12}/255$ | $\epsilon = {}^{16}/255$ |
|---|---|---|---|---|
| ATAS | $90.63 \pm 0.26$ $61.53 \pm 0.17$ | $87.25 \pm 0.13$ $41.91 \pm 0.38$ | $86.89 \pm 1.83$ $12.08 \pm 9.60$ | $83.89 \pm 1.45$ $8.86 \pm 6.29$ |
| MultiGrad | $87.56 \pm 0.07$ $66.49 \pm 0.22$ | $80.62 \pm 0.22$ $43.33 \pm 0.25$ | $73.50 \pm 0.36$ $36.65 \pm 0.63$ | $74.87 \pm 3.70$ $0.00 \pm 0.00$ |
| NFGSM | $87.40 \pm 0.18$ $66.44 \pm 0.20$ | $79.05 \pm 0.21$ $48.73 \pm 0.44$ | $69.36 \pm 0.35$ $37.54 \pm 0.63$ | $61.20 \pm 0.57$ $29.23 \pm 0.59$ |
| SORA (Ours) | $87.68 \pm 0.28$ $66.65 \pm 0.20$ | $79.95 \pm 0.01$ $48.72 \pm 0.30$ | $71.36 \pm 0.14$ $37.63 \pm 0.40$ | $63.01 \pm 0.56$ $29.13 \pm 0.51$ |

Table 6: Training results for PreActResNet-18 on the CIFAR-10. The best results are shown in **bold** and the second best results are underlined. Methods which outperform SORA are shown in green and those that perform worse are shown in red.

| Method | $\epsilon = {}^{2}/255$ | | $\epsilon = {}^{4}/255$ | | $\epsilon = {}^{8}/255$ | | $\epsilon = {}^{12}/255$ | | $\epsilon = {}^{16}/255$ | |
|---|---|---|---|---|---|---|---|---|---|---|
| | FGSM | PGD | FGSM | PGD | FGSM | PGD | FGSM | PGD | FGSM | PGD |
| FGSM | 38.17 | 0.19 | **91.33** | 0.04 | **58.07** | 0.04 | **49.33** | 0.07 | **41.18** | 0.04 |
| FGSM-RS | 72.47 | 71.76 | 64.77 | **62.72** | 49.29 | 41.89 | 38.17 | 24.40 | 30.17 | 10.64 |
| GradAlign | 72.02 | 71.61 | 63.84 | 61.64 | 49.74 | 42.04 | 37.91 | 23.44 | 29.65 | 10.49 |
| ZeroGrad | 71.35 | 70.72 | 63.88 | 61.98 | 50.37 | 43.75 | 40.40 | 25.82 | 31.73 | 12.50 |
| MultiGrad | 70.50 | 70.05 | 63.24 | 61.53 | 50.63 | 43.64 | 39.96 | 26.56 | 31.36 | 12.72 |
| NFGSM | 69.31 | 69.08 | 62.83 | 61.35 | 50.26 | 44.31 | 39.73 | 27.94 | 31.36 | 14.21 |
| ATAS | **75.37** | **74.26** | 65.40 | 60.83 | 47.92 | 36.61 | 35.16 | 16.59 | 26.19 | 6.21 |
| AAER | 69.49 | 69.35 | 63.02 | 61.42 | 50.60 | **44.64** | 40.10 | **28.31** | 31.70 | 14.06 |
| ELLE | 72.14 | 71.35 | 64.69 | 61.90 | 48.51 | 39.69 | 36.05 | 20.09 | 26.08 | 7.96 |
| **SORA** | 70.57 | 69.90 | 63.17 | 61.76 | 50.74 | 44.46 | 40.10 | 27.19 | 32.07 | **14.36** |
| PGD | **70.87** | **70.50** | **64.14** | **63.10** | **50.86** | **45.94** | **40.66** | 30.32 | 31.47 | 16.96 |
| TRADES | 69.46 | 69.05 | 62.50 | 61.16 | 50.60 | 45.87 | 40.36 | **32.59** | **32.51** | **20.05** |

## F.5 EPSILON OVERFITTING

We demonstrate the occurrence of Epsilon Overfitting (EO) under varying training perturbation magnitudes and across different datasets.

Varying the training value of $\epsilon$ influences the peak of FGSM accuracy following CO Figure 9. Larger $\epsilon$ values shift these peaks toward higher perturbation magnitudes, whereas smaller values produce sharper peaks.

EO is also evident across multiple datasets, as shown in Figure 10.

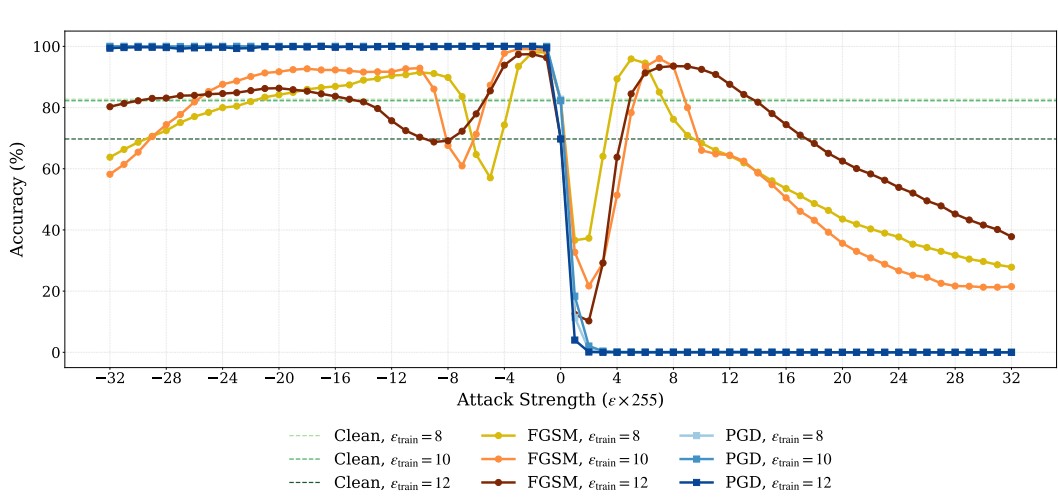

Figure 9: Overlay of FGSM accuracies showing the effect of different training $\epsilon$ values on the EO peak location and sharpness.

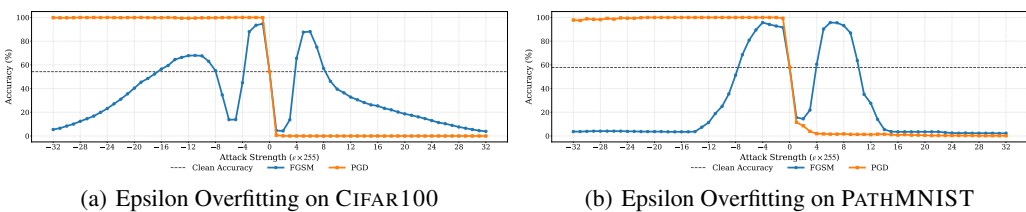

(a) Epsilon Overfitting on CIFAR100

(b) Epsilon Overfitting on PATHMNIST

Figure 10: Examples of EO occurrence across datasets with models exhibiting CO.

## G    GENERALIZABILITY OF PERTALIGN

To demonstrate the predictive power of the PertAlign metric, we present training traces from various architectures, methods, and datasets. The correlation between catastrophic overfitting (CO) and concurrent drops in PGD robustness and PertAlign is clearly evident. For instance, Figure 11(g) shows PertAlign dropping, recovering, and dropping again as the model undergoes CO, temporarily recovers, and then succumbs to CO once more.

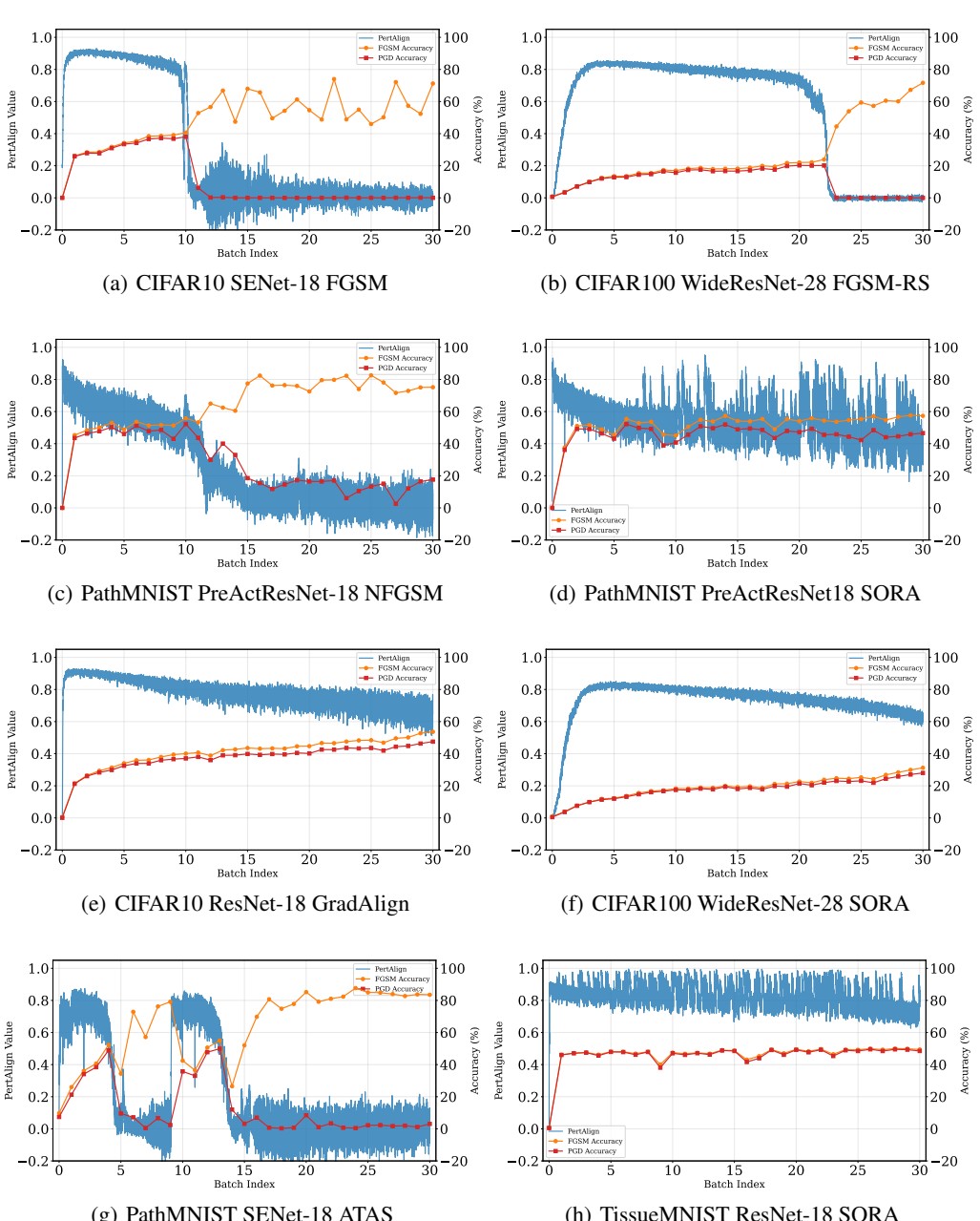

(a) CIFAR10 SENet-18 FGSM

(b) CIFAR100 WideResNet-28 FGSM-RS

(c) PathMNIST PreActResNet-18 NFGSM

(d) PathMNIST PreActResNet18 SORA

(e) CIFAR10 ResNet-18 GradAlign

(f) CIFAR100 WideResNet-28 SORA

(g) PathMNIST SENet-18 ATAS

(h) TissueMNIST ResNet-18 SORA

Figure 11: Tracking PertAlign across datasets, models, and FAT methods.

More formally, the relationship between catastrophic overfitting and PertAlign can be explained by considering the loss landscape. During normal adversarial training, moving along the gradient ascent direction should increase the loss. However, during CO, we observe a decline in loss values after a certain perturbation threshold, indicating that the loss begins to decrease beyond this point.

This behavior *requires* high curvature in the gradient direction; otherwise, the loss would increase monotonically along the gradient path.

From an optimization perspective, multi-step gradient ascent with appropriate step sizes could mitigate this issue by carefully navigating the loss landscape. However, in single-step adversarial attacks, as in fast adversarial training, the step size becomes the critical parameter. SORA addresses this by using the PertAlign metric to estimate the local curvature and adaptively adjust the step size, avoiding overshooting that leads to CO.

Overshooting along the gradient direction is problematic not only because it generates weaker or even abnormal adversarial examples, but also because training on these examples can distort the loss surface further, as illustrated in Figure 4. This distortion may reduce margins, defined as the distance from inputs to the decision boundary, contrary to the goal of improving robust accuracy (Ding et al., 2020; Sriramanan et al., 2020).

# H EXTENDED RESULTS

We provide detailed results of our experiments on methods, datasets and architectures from Appendices B, D and E. Figure 5 presents an overview of these results. The relative accuracy reported in Figure 5 is obtained via dividing accuracies for each setting by the highest accuracy in that setting. This allows us to analyze the results across a wide range of settings in a single figure.

In this appendix, we report the experimental results for every dataset–architecture pair used in our study. In all tables, the best results are shown in **bold** and the second best results have been underlined.

ATAS exhibits significantly higher memory consumption compared to other methods, as illustrated by its resource usage on CIFAR-10 with PreActResNet-18 in Figure 6(b). This substantial memory requirement prevented us from running ATAS on TINYIMAGENET using our NVIDIA RTX 4090 GPU due to hardware limitations.

Rocamora et al. (2024) recommend $\lambda$ values in the range $[4000, 20000]$ to avoid Catastrophic Overfitting (CO) and achieve optimal performance. However, we found that for WideResNet architectures, this parameter range, combined with the typically large values of their regularizer (e.g., $10^{20}$), frequently leads to numerical overflow. This instability causes the loss regularizer term to become NaN, ultimately disrupting the training process.

Table 7: **CIFAR10 Dataset - PreActResNet-18**

| Method | Clean | FGSM | PGD-10 | AA |
|--------|-------|------|--------|-----|
| FGSM | 83.37 ± 0.71 | **66.00** ± 7.98 | 0.02 ± 0.02 | 0.00 ± 0.00 |
| FGSM-RS | 82.61 ± 0.03 | 53.32 ± 0.36 | 47.16 ± 0.35 | 42.48 ± 0.23 |
| GradAlign | 82.40 ± 0.17 | 53.11 ± 0.35 | 47.02 ± 0.48 | 42.49 ± 0.39 |
| ZeroGrad | 80.92 ± 0.04 | 53.52 ± 0.27 | 47.84 ± 0.38 | 43.05 ± 0.23 |
| MultiGrad | 80.64 ± 0.22 | 53.76 ± 0.24 | 48.22 ± 0.27 | 43.33 ± 0.25 |
| NFGSM | 79.05 ± 0.21 | 53.32 ± 0.22 | 48.73 ± 0.44 | 43.95 ± 0.23 |
| ATAS | **87.25** ± 0.13 | 51.97 ± 0.13 | 41.91 ± 0.38 | 37.25 ± 0.40 |
| AAER | 79.07 ± 0.21 | 53.44 ± 0.06 | **48.80** ± 0.38 | 43.96 ± 0.26 |
| ELLE | 83.74 ± 0.51 | 52.59 ± 0.18 | 45.10 ± 0.35 | 40.63 ± 0.25 |
| SORA (Ours) | 79.95 ± 0.01 | 53.74 ± 0.14 | 48.72 ± 0.30 | **44.00** ± 0.13 |
| PGD-10 | **79.37** ± 0.12 | **54.38** ± 0.13 | 50.60 ± 0.10 | 46.15 ± 0.19 |
| TRADES | 78.45 ± 0.42 | 54.16 ± 0.13 | **50.88** ± 0.10 | **46.50** ± 0.10 |

Table 8: **CIFAR10 Dataset - ResNet-18**

| Method | Clean | FGSM | PGD-10 | AA |
|--------|-------|------|--------|-----|
| FGSM | 84.09 ± 1.85 | **85.32** ± 7.82 | 0.09 ± 0.06 | 0.00 ± 0.00 |
| FGSM-RS | 78.90 ± 5.28 | 66.18 ± 13.80 | 15.95 ± 21.98 | 14.15 ± 20.01 |
| GradAlign | 82.40 ± 0.22 | 53.70 ± 0.15 | 47.29 ± 0.21 | 42.49 ± 0.05 |
| ZeroGrad | 80.74 ± 0.15 | 54.11 ± 0.12 | 47.94 ± 0.54 | 42.61 ± 0.94 |
| MultiGrad | 80.66 ± 0.14 | 53.99 ± 0.05 | 48.45 ± 0.29 | 43.33 ± 0.14 |
| NFGSM | 78.91 ± 0.30 | 53.52 ± 0.30 | 48.76 ± 0.43 | 43.96 ± 0.31 |
| ATAS | **87.23** ± 0.24 | 52.44 ± 0.22 | 42.43 ± 0.18 | 38.06 ± 0.31 |
| AAER | 78.91 ± 0.30 | 53.52 ± 0.30 | 48.76 ± 0.43 | 43.96 ± 0.32 |
| ELLE | 83.30 ± 0.13 | 52.21 ± 0.23 | 45.02 ± 0.30 | 40.33 ± 0.13 |
| SORA (Ours) | 79.81 ± 0.20 | 53.89 ± 0.32 | **49.00** ± 0.36 | **44.06** ± 0.51 |
| PGD-10 | **79.17** ± 0.03 | **54.62** ± 0.14 | **50.99** ± 0.23 | **46.46** ± 0.17 |
| TRADES | 78.19 ± 0.16 | 53.89 ± 0.09 | 50.66 ± 0.11 | 46.42 ± 0.16 |

Table 9: **CIFAR10 Dataset - WideResNet-28**

| Method | Clean | FGSM | PGD-10 | AA |
|---|---|---|---|---|
| FGSM | 86.20 ± 0.58 | 80.51 ± 2.38 | 0.09 ± 0.03 | 0.00 ± 0.00 |
| FGSM-RS | 84.12 ± 2.40 | **92.10** ± 3.47 | 0.06 ± 0.07 | 0.00 ± 0.00 |
| GradAlign | 85.72 ± 0.42 | 55.42 ± 0.25 | 47.92 ± 0.10 | 43.63 ± 0.34 |
| ZeroGrad | 81.66 ± 4.07 | 87.39 ± 6.19 | 0.01 ± 0.02 | 0.00 ± 0.00 |
| MultiGrad | 84.92 ± 0.30 | 55.98 ± 0.09 | 48.60 ± 0.21 | 43.85 ± 0.25 |
| NFGSM | 82.54 ± 0.35 | 55.26 ± 0.13 | 49.48 ± 0.08 | 45.26 ± 0.19 |
| ATAS | **89.07** ± 0.12 | 54.58 ± 0.43 | 44.11 ± 0.45 | 39.52 ± 0.34 |
| AAER | 82.54 ± 0.35 | 55.26 ± 0.13 | 49.48 ± 0.08 | **45.27** ± 0.19 |
| ELLE | 10.00 ± 0.00 | 10.00 ± 0.00 | 10.00 ± 0.00 | 10.00 ± 0.00 |
| SORA (Ours) | 83.57 ± 0.40 | 55.80 ± 0.22 | **49.53** ± 0.17 | 45.17 ± 0.08 |
| PGD-10 | **83.34** ± 0.67 | **57.04** ± 0.46 | **51.89** ± 0.56 | 47.17 ± 0.61 |
| TRADES | 60.67 ± 27.95 | 43.13 ± 17.92 | 40.69 ± 16.21 | **47.77** ± 0.16 |

Table 10: **CIFAR10 Dataset - SENet-18**

| Method | Clean | FGSM | PGD-10 | AA |
|---|---|---|---|---|
| FGSM | 84.14 ± 1.17 | **76.74** ± 13.43 | 0.04 ± 0.04 | 0.00 ± 0.00 |
| FGSM-RS | 77.91 ± 6.72 | 59.19 ± 8.19 | 31.00 ± 21.87 | 27.49 ± 19.44 |
| GradAlign | 82.75 ± 0.39 | 53.75 ± 0.09 | 47.26 ± 0.24 | 42.41 ± 0.21 |
| ZeroGrad | 81.83 ± 3.59 | 75.40 ± 17.00 | 15.96 ± 22.57 | 14.32 ± 20.26 |
| MultiGrad | 81.30 ± 0.51 | 54.20 ± 0.38 | 48.62 ± 0.19 | 43.61 ± 0.25 |
| NFGSM | 79.44 ± 0.39 | 53.63 ± 0.22 | **48.91** ± 0.12 | 43.93 ± 0.23 |
| ATAS | **87.38** ± 0.07 | 52.37 ± 0.39 | 41.80 ± 0.15 | 37.25 ± 0.16 |
| AAER | 79.44 ± 0.39 | 53.63 ± 0.22 | **48.91** ± 0.12 | 43.93 ± 0.22 |
| ELLE | 83.68 ± 0.28 | 52.64 ± 0.13 | 45.44 ± 0.19 | 40.86 ± 0.21 |
| SORA (Ours) | 80.19 ± 0.29 | 53.81 ± 0.34 | 48.90 ± 0.20 | **43.98** ± 0.36 |
| PGD-10 | **79.62** ± 0.09 | **54.93** ± 0.13 | **51.27** ± 0.18 | 46.42 ± 0.47 |
| TRADES | 78.66 ± 0.41 | 54.01 ± 0.10 | 50.90 ± 0.27 | **46.54** ± 0.34 |

Table 11: **CIFAR100 Dataset - PreActResNet-18**

| Method | Clean | FGSM | PGD-10 | AA |
|---|---|---|---|---|
| FGSM | 52.75 ± 2.06 | 54.55 ± 26.45 | 0.01 ± 0.01 | 0.00 ± 0.00 |
| FGSM-RS | 52.70 ± 0.52 | **55.02** ± 2.45 | 0.00 ± 0.00 | 0.00 ± 0.00 |
| GradAlign | 57.50 ± 0.85 | 28.62 ± 0.11 | 25.20 ± 0.30 | 20.67 ± 0.18 |
| ZeroGrad | 56.28 ± 0.47 | 28.37 ± 0.04 | 25.32 ± 0.13 | 20.81 ± 0.28 |
| MultiGrad | 55.53 ± 0.55 | 28.98 ± 0.27 | 26.09 ± 0.21 | 21.45 ± 0.05 |
| NFGSM | 53.43 ± 0.40 | 28.82 ± 0.12 | 26.25 ± 0.18 | **21.75** ± 0.18 |
| ATAS | **63.57** ± 0.17 | 27.34 ± 0.30 | 21.10 ± 0.19 | 17.17 ± 0.16 |
| AAER | 53.43 ± 0.40 | 28.82 ± 0.12 | 26.25 ± 0.18 | 21.72 ± 0.17 |
| ELLE | 57.94 ± 0.47 | 27.74 ± 0.46 | 23.79 ± 0.28 | 19.70 ± 0.05 |
| SORA (Ours) | 54.65 ± 0.73 | 29.08 ± 0.20 | **26.33** ± 0.05 | 21.71 ± 0.11 |
| PGD-10 | 53.57 ± 0.69 | 29.66 ± 0.11 | 27.64 ± 0.06 | **22.88** ± 0.06 |
| TRADES | **55.42** ± 0.58 | **29.86** ± 0.25 | **28.05** ± 0.23 | 22.77 ± 0.08 |

Table 12: **CIFAR100 Dataset - ResNet-18**

| Method | Clean | FGSM | PGD-10 | AA |
|---|---|---|---|---|
| FGSM | 54.17 ± 1.97 | 49.22 ± 22.81 | 0.01 ± 0.00 | 0.00 ± 0.00 |
| FGSM-RS | 50.51 ± 5.11 | **66.70** ± 9.54 | 0.00 ± 0.00 | 0.00 ± 0.00 |
| GradAlign | 57.24 ± 0.41 | 28.85 ± 0.01 | 25.37 ± 0.13 | 20.97 ± 0.17 |
| ZeroGrad | 56.22 ± 0.51 | 28.60 ± 0.17 | 25.49 ± 0.15 | 20.96 ± 0.14 |
| MultiGrad | 55.48 ± 0.49 | 29.17 ± 0.13 | 26.17 ± 0.03 | 21.63 ± 0.10 |
| NFGSM | 53.26 ± 0.66 | 28.59 ± 0.10 | 26.21 ± 0.05 | 21.81 ± 0.06 |
| ATAS | **63.39** ± 0.16 | 27.71 ± 0.64 | 21.68 ± 0.31 | 17.50 ± 0.26 |
| AAER | 53.26 ± 0.66 | 28.59 ± 0.10 | 26.21 ± 0.05 | 21.85 ± 0.07 |
| ELLE | 57.53 ± 0.41 | 27.68 ± 0.25 | 23.94 ± 0.48 | 19.61 ± 0.46 |
| SORA (Ours) | 54.56 ± 0.18 | 29.19 ± 0.08 | **26.61** ± 0.19 | **21.99** ± 0.30 |
| PGD-10 | 53.39 ± 0.40 | 29.68 ± 0.29 | 27.61 ± 0.41 | **22.93** ± 0.19 |
| TRADES | **55.00** ± 0.59 | **30.13** ± 0.19 | **28.16** ± 0.23 | 22.79 ± 0.07 |

Table 13: **CIFAR100 Dataset - WideResNet-28**

| Method | Clean | FGSM | PGD-10 | AA |
|---|---|---|---|---|
| FGSM | 57.74 ± 1.07 | 41.99 ± 10.99 | 0.06 ± 0.04 | 0.00 ± 0.00 |
| FGSM-RS | 48.19 ± 1.36 | **70.83** ± 0.84 | 0.00 ± 0.00 | 0.00 ± 0.00 |
| GradAlign | 60.43 ± 1.46 | 41.21 ± 8.98 | 9.96 ± 12.31 | 7.74 ± 10.83 |
| ZeroGrad | 50.57 ± 5.73 | 69.10 ± 4.99 | 1.07 ± 1.50 | 0.09 ± 0.13 |
| MultiGrad | 60.07 ± 0.26 | 31.02 ± 0.03 | 27.37 ± 0.08 | 23.08 ± 0.06 |
| NFGSM | 57.28 ± 0.19 | 30.50 ± 0.34 | 27.82 ± 0.38 | 23.74 ± 0.29 |
| ATAS | **68.18** ± 0.88 | 40.85 ± 3.88 | 16.07 ± 2.36 | 7.21 ± 3.27 |
| AAER | 57.28 ± 0.19 | 30.50 ± 0.34 | 27.82 ± 0.38 | 23.75 ± 0.29 |
| ELLE | 01.00 ± 0.00 | 01.00 ± 0.01 | 0.00 ± 0.00 | 01.00 ± 0.00 |
| SORA (Ours) | 58.30 ± 0.17 | 31.22 ± 0.12 | **28.02** ± 0.15 | **23.79** ± 0.24 |
| PGD-10 | **57.97** ± 0.24 | **32.26** ± 0.20 | 29.57 ± 0.32 | **25.20** ± 0.28 |
| TRADES | 57.76 ± 0.02 | 32.17 ± 0.31 | **30.15** ± 0.35 | 24.85 ± 0.24 |

Table 14: **CIFAR100 Dataset - SENet-18**

| Method | Clean | FGSM | PGD-10 | AA |
|---|---|---|---|---|
| FGSM | 50.06 ± 2.31 | 37.93 ± 6.81 | 0.16 ± 0.05 | 0.00 ± 0.00 |
| FGSM-RS | 49.62 ± 9.38 | **44.65** ± 21.85 | 7.87 ± 11.11 | 5.71 ± 8.08 |
| GradAlign | 56.44 ± 0.51 | 28.34 ± 0.30 | 25.06 ± 0.14 | 20.86 ± 0.01 |
| ZeroGrad | 55.24 ± 0.39 | 28.21 ± 0.34 | 25.11 ± 0.11 | 20.62 ± 0.00 |
| MultiGrad | 54.36 ± 0.24 | 28.79 ± 0.34 | 25.86 ± 0.22 | 21.31 ± 0.04 |
| NFGSM | 52.30 ± 0.24 | 28.27 ± 0.39 | 25.85 ± 0.16 | 21.55 ± 0.02 |
| ATAS | **62.48** ± 0.23 | 27.12 ± 0.13 | 21.31 ± 0.21 | 17.69 ± 0.26 |
| AAER | 52.30 ± 0.24 | 28.27 ± 0.39 | 25.85 ± 0.16 | **21.59** ± 0.00 |
| ELLE | 57.04 ± 0.13 | 27.35 ± 0.15 | 23.70 ± 0.18 | 19.42 ± 0.06 |
| SORA (Ours) | 53.67 ± 0.11 | 28.85 ± 0.58 | **26.42** ± 0.39 | 21.57 ± 0.07 |
| PGD-10 | 52.52 ± 0.39 | 29.11 ± 0.27 | 26.94 ± 0.13 | 22.44 ± 0.00 |
| TRADES | **54.11** ± 0.37 | **29.41** ± 0.13 | **27.49** ± 0.06 | **22.52** ± 0.00 |

Table 15: **TinyImageNet Dataset - PreActNet-18**

| Method | Clean | FGSM | PGD-10 |
|--------|-------|------|--------|
| FGSM | 36.61 ± 0.10 | **88.21** ± 3.54 | 0.01 ± 0.01 |
| FGSM-RS | 47.46 ± 0.11 | 22.31 ± 0.11 | 19.67 ± 0.22 |
| GradAlign | 47.67 ± 0.00 | 21.86 ± 0.00 | 19.32 ± 0.00 |
| ZeroGrad | 46.00 ± 0.07 | 21.56 ± 0.10 | 19.28 ± 0.06 |
| MultiGrad | 45.42 ± 0.11 | 22.27 ± 0.15 | 19.98 ± 0.16 |
| NFGSM | 44.66 ± 0.20 | 22.14 ± 0.28 | **20.18** ± 0.18 |
| AAER | 44.72 ± 0.25 | 22.02 ± 0.12 | 19.94 ± 0.25 |
| ELLE | **48.80** ± 0.03 | 21.61 ± 0.11 | 18.69 ± 0.04 |
| SORA (Ours) | 44.98 ± 0.46 | 22.03 ± 0.09 | 19.97 ± 0.05 |

Table 16: **TinyImageNet Dataset - ResNet-18**

| Method | Clean | FGSM | PGD-10 |
|--------|-------|------|--------|
| FGSM | 39.10 ± 0.60 | **38.16** ± 14.90 | 5.75 ± 8.13 |
| FGSM-RS | 45.92 ± 4.16 | 17.59 ± 7.37 | 0.05 ± 0.07 |
| GradAlign | 48.49 ± 0.00 | 22.44 ± 0.00 | 19.74 ± 0.00 |
| ZeroGrad | 46.75 ± 0.02 | 22.09 ± 0.12 | 19.36 ± 0.07 |
| MultiGrad | 46.24 ± 0.23 | 22.69 ± 0.15 | 20.26 ± 0.18 |
| NFGSM | 45.19 ± 0.08 | 22.36 ± 0.20 | 20.16 ± 0.29 |
| AAER | 45.11 ± 0.39 | 22.45 ± 0.20 | **20.29** ± 0.18 |
| ELLE | **49.43** ± 0.17 | 21.67 ± 0.25 | 18.56 ± 0.18 |
| SORA (Ours) | 45.60 ± 0.16 | 22.40 ± 0.04 | 19.99 ± 0.06 |

Table 17: **TinyImageNet Dataset - SENet-18**

| Method | Clean | FGSM | PGD-10 |
|--------|-------|------|--------|
| FGSM | 42.82 ± 0.69 | 20.81 ± 0.46 | 18.94 ± 0.30 |
| FGSM-RS | 46.41 ± 0.22 | 21.57 ± 0.12 | 19.02 ± 0.04 |
| GradAlign | 47.02 ± 0.00 | 21.72 ± 0.00 | 19.42 ± 0.00 |
| ZeroGrad | 45.39 ± 0.48 | 21.39 ± 0.12 | 19.18 ± 0.00 |
| MultiGrad | 45.03 ± 0.39 | **22.03** ± 0.27 | 19.80 ± 0.16 |
| NFGSM | 43.20 ± 0.59 | 21.79 ± 0.02 | 20.06 ± 0.13 |
| AAER | 43.48 ± 0.07 | 21.66 ± 0.14 | 19.86 ± 0.31 |
| ELLE | **48.42** ± 0.14 | 21.26 ± 0.12 | 18.41 ± 0.13 |
| SORA (Ours) | 44.28 ± 0.07 | **21.97** ± 0.24 | **20.16** ± 0.31 |

Table 18: **PathMNIST Dataset - PreActResNet-18**

| Method | Clean | FGSM | PGD-10 | AA |
|---|---|---|---|---|
| FGSM | 46.06 ± 9.99 | 84.62 ± 6.84 | 2.79 ± 0.82 | 0.42 ± 0.16 |
| FGSM-RS | 60.34 ± 8.01 | **86.96** ± 4.37 | 1.72 ± 0.95 | 1.44 ± 0.44 |
| GradAlign | 45.71 ± 13.74 | 38.91 ± 25.41 | 0.78 ± 1.10 | 0.77 ± 1.09 |
| ZeroGrad | 67.22 ± 14.19 | 73.85 ± 13.12 | 5.69 ± 3.45 | 1.05 ± 0.58 |
| MultiGrad | 82.11 ± 3.26 | 56.90 ± 6.03 | 37.29 ± 4.88 | 24.94 ± 5.40 |
| NFGSM | 74.86 ± 6.14 | 55.39 ± 14.43 | 18.84 ± 4.14 | 1.61 ± 0.79 |
| ATAS | 62.50 ± 10.88 | 71.48 ± 5.83 | 1.71 ± 0.58 | 1.08 ± 0.82 |
| AAER | 74.86 ± 6.14 | 55.39 ± 14.43 | 18.84 ± 4.14 | 1.61 ± 0.79 |
| ELLE | 79.80 ± 6.67 | 50.16 ± 6.19 | 40.52 ± 2.38 | **34.60** ± 1.05 |
| SORA (Ours) | **85.93** ± 1.95 | 55.41 ± 1.38 | **41.13** ± 4.23 | 32.91 ± 4.87 |
| PGD-10 | **77.62** ± 1.45 | **57.63** ± 1.22 | **54.47** ± 0.89 | **50.70** ± 0.53 |
| TRADES | 75.25 ± 0.73 | 54.75 ± 0.86 | 51.94 ± 0.88 | 48.28 ± 0.74 |

Table 19: **PathMNIST Dataset - ResNet-18**

| Method | Clean | FGSM | PGD-10 | AA |
|---|---|---|---|---|
| FGSM | 36.69 ± 2.62 | 79.98 ± 3.99 | 0.63 ± 0.15 | 0.30 ± 0.20 |
| FGSM-RS | 30.70 ± 7.65 | **87.13** ± 2.48 | 2.05 ± 0.21 | 1.50 ± 0.41 |
| GradAlign | 32.62 ± 4.94 | 38.07 ± 32.26 | 2.60 ± 2.11 | 0.72 ± 1.02 |
| ZeroGrad | 55.37 ± 17.08 | 60.24 ± 21.54 | 2.66 ± 0.50 | 1.28 ± 0.72 |
| MultiGrad | 81.88 ± 1.15 | 57.89 ± 2.96 | 38.19 ± 4.10 | 29.15 ± 4.48 |
| NFGSM | 71.90 ± 12.82 | 86.18 ± 5.95 | 14.74 ± 10.11 | 9.15 ± 6.46 |
| ATAS | 66.49 ± 4.94 | 70.95 ± 8.06 | 1.56 ± 0.75 | 0.68 ± 0.34 |
| AAER | 71.90 ± 12.82 | 86.18 ± 5.95 | 14.74 ± 10.11 | 9.16 ± 6.48 |
| ELLE | 79.65 ± 6.09 | 50.49 ± 4.63 | 41.75 ± 1.38 | 36.22 ± 0.49 |
| SORA (Ours) | **84.01** ± 0.60 | 56.18 ± 1.62 | **45.36** ± 2.11 | **38.11** ± 1.94 |
| PGD-10 | **76.63** ± 1.66 | **55.51** ± 2.21 | **52.13** ± 1.99 | **48.41** ± 1.31 |
| TRADES | 18.64 ± 0.00 | 18.64 ± 0.00 | 18.64 ± 0.00 | 18.64 ± 0.00 |

Table 20: **PathMNIST Dataset - WideResNet-28**

| Method | Clean | FGSM | PGD-10 | AA |
|---|---|---|---|---|
| FGSM | 48.43 ± 9.79 | 78.60 ± 0.91 | 2.31 ± 0.45 | 1.33 ± 0.42 |
| FGSM-RS | 32.01 ± 8.07 | 86.52 ± 2.67 | 2.04 ± 0.33 | 1.28 ± 0.69 |
| GradAlign | 39.37 ± 11.13 | 65.33 ± 8.27 | 1.98 ± 0.54 | 1.69 ± 0.69 |
| ZeroGrad | 51.96 ± 19.24 | 85.71 ± 3.27 | 3.24 ± 1.02 | 2.12 ± 0.17 |
| MultiGrad | **85.79** ± 1.22 | 59.65 ± 0.94 | 40.97 ± 2.24 | 29.48 ± 2.50 |
| NFGSM | 84.01 ± 3.06 | **90.80** ± 5.19 | 16.34 ± 8.90 | 8.77 ± 6.44 |
| ATAS | 55.53 ± 9.27 | 79.78 ± 5.60 | 2.34 ± 0.00 | 2.08 ± 0.26 |
| AAER | 84.01 ± 3.06 | **90.80** ± 5.19 | 16.34 ± 8.90 | 12.46 ± 4.55 |
| ELLE | 18.64 ± 0.00 | 18.64 ± 0.00 | 18.64 ± 0.00 | 18.64 ± 0.00 |
| SORA (Ours) | 84.34 ± 0.81 | 56.36 ± 1.52 | **45.32** ± 2.04 | **38.20** ± 2.89 |
| PGD-10 | **79.16** ± 1.11 | **58.68** ± 0.34 | **53.74** ± 0.41 | 48.86 ± 0.00 |
| TRADES | 56.96 ± 27.11 | 43.20 ± 17.37 | 41.15 ± 15.92 | **32.30** ± 13.66 |

Table 21: **PathMNIST Dataset - SENet-18**

| Method | Clean | FGSM | PGD-10 |
|---|---|---|---|
| FGSM | 52.42 ± 6.18 | **89.58** ± 1.98 | 2.49 ± 2.15 |
| FGSM-RS | 54.88 ± 14.63 | 89.31 ± 3.15 | 2.92 ± 2.47 |
| GradAlign | 41.02 ± 14.03 | 65.05 ± 4.76 | 0.86 ± 0.87 |
| ZeroGrad | 60.06 ± 9.01 | 83.25 ± 5.65 | 1.49 ± 0.75 |
| MultiGrad | 73.65 ± 1.64 | 49.05 ± 3.58 | 31.91 ± 0.74 |
| NFGSM | 79.33 ± 1.78 | 77.91 ± 15.64 | 23.99 ± 15.29 |
| ATAS | 38.22 ± 16.22 | 81.83 ± 1.26 | 7.09 ± 7.50 |
| AAER | 79.33 ± 1.78 | 77.91 ± 15.64 | 23.99 ± 15.29 |
| ELLE | 76.45 ± 5.45 | 49.62 ± 2.31 | 44.10 ± 1.36 |
| SORA (Ours) | **82.33** ± 0.53 | 58.12 ± 1.19 | **47.74** ± 0.66 |
| PGD-10 | **76.23** ± 1.67 | **53.72** ± 2.39 | **50.50** ± 2.13 |
| TRADES | 18.64 ± 0.00 | 18.64 ± 0.00 | 18.64 ± 0.00 |

Table 22: **TissueMNIST Dataset - PreActResNet-18**

| Method | Clean | FGSM | PGD-10 |
|---|---|---|---|
| FGSM | 51.80 ± 2.75 | **81.39** ± 12.14 | 0.28 ± 0.11 |
| FGSM-RS | 52.32 ± 4.00 | 80.82 ± 3.75 | 0.35 ± 0.16 |
| GradAlign | 55.41 ± 2.72 | 70.13 ± 15.40 | 16.23 ± 22.48 |
| ZeroGrad | 57.36 ± 1.65 | 69.01 ± 2.22 | 0.10 ± 0.01 |
| MultiGrad | **61.79** ± 2.54 | 42.72 ± 4.99 | 16.92 ± 22.61 |
| NFGSM | 57.47 ± 0.10 | 49.75 ± 0.02 | 49.25 ± 0.06 |
| ATAS | 31.95 ± 10.54 | 56.36 ± 6.04 | 0.08 ± 0.06 |
| AAER | 57.47 ± 0.10 | 49.75 ± 0.02 | 49.25 ± 0.06 |
| ELLE | 60.57 ± 0.48 | 47.61 ± 0.67 | 46.07 ± 0.92 |
| SORA (Ours) | 58.06 ± 0.22 | 50.05 ± 0.10 | **49.31** ± 0.05 |
| PGD-10 | **57.75** ± 0.09 | **50.13** ± 0.04 | **49.62** ± 0.09 |

Table 23: **TissueMNIST Dataset - ResNet-18**

| Method | Clean | FGSM | PGD-10 |
|---|---|---|---|
| FGSM | 50.54 ± 0.16 | 83.73 ± 5.25 | 0.26 ± 0.14 |
| FGSM-RS | 53.06 ± 1.58 | **85.55** ± 3.23 | 0.24 ± 0.22 |
| GradAlign | 48.24 ± 4.25 | 58.93 ± 14.77 | 5.90 ± 7.58 |
| ZeroGrad | 55.70 ± 1.05 | 67.73 ± 2.22 | 0.14 ± 0.07 |
| MultiGrad | 58.28 ± 0.30 | 50.34 ± 0.21 | 49.22 ± 0.06 |
| NFGSM | 57.38 ± 0.18 | 49.97 ± 0.03 | **49.51** ± 0.05 |
| ATAS | 46.32 ± 18.21 | 42.74 ± 4.30 | 29.77 ± 21.03 |
| AAER | 57.38 ± 0.18 | 49.97 ± 0.03 | **49.51** ± 0.05 |
| ELLE | **59.18** ± 0.24 | 49.20 ± 0.11 | 48.43 ± 0.16 |
| SORA (Ours) | 58.63 ± 0.77 | 49.78 ± 0.25 | 49.09 ± 0.44 |
| PGD-10 | **57.62** ± 0.18 | **50.23** ± 0.08 | **49.76** ± 0.09 |

Table 24: **TissueMNIST Dataset - SENet-18**

| Method | Clean | FGSM | PGD-10 |
|---|---|---|---|
| FGSM | 47.26 ± 1.51 | 84.39 ± 2.67 | 0.26 ± 0.10 |
| FGSM-RS | 56.10 ± 0.98 | **86.98** ± 3.53 | 0.44 ± 0.21 |
| GradAlign | 52.50 ± 2.08 | 84.80 ± 5.94 | 0.02 ± 0.03 |
| ZeroGrad | 56.23 ± 0.91 | 70.19 ± 1.10 | 0.06 ± 0.03 |
| MultiGrad | 59.25 ± 1.65 | 48.02 ± 2.99 | 32.96 ± 22.97 |
| NFGSM | 57.21 ± 0.05 | 50.01 ± 0.20 | 49.58 ± 0.20 |
| ATAS | 48.39 ± 11.60 | 51.68 ± 5.24 | 18.15 ± 17.66 |
| AAER | 57.21 ± 0.05 | 50.01 ± 0.20 | 49.58 ± 0.20 |
| ELLE | **60.09** ± 0.47 | 48.21 ± 0.51 | 47.22 ± 0.69 |
| SORA (Ours) | 57.87 ± 0.20 | 50.25 ± 0.03 | **49.63** ± 0.16 |
| PGD-10 | **57.19** ± 0.30 | **50.15** ± 0.12 | **49.73** ± 0.11 |

# I   Research Statement

## I.1   Reproducibility

### I.1.1   Main Results

We have made our code and experimental results publicly available[1] to ensure full reproducibility. General hyperparameters and optimizer details are provided in Section 6.1. As recommended by the original authors of the baseline models, we provide a comprehensive account of their hyperparameters and the rationale for their selection in Appendix B. Dataset details and corresponding augmentation strategies are also documented in Appendix D.

All experimental configurations, including random seeds, are accessible in our code repository. Due to computational constraints, we conducted experiments using three distinct random seeds to assess the stability of our results. The detailed outcomes for each seed are available in Appendix H.

### I.1.2   Figures

The code for generating all figures is included in our repository. The data for Figure 2 and Figure 9 were produced by our main codebase. For visual clarity, these figures were subsequently generated by processing and combining this raw data. We emphasize that the co-occurrence of Catastrophic Overfitting with drops in PertAlign (and GradAlign) to zero was consistently observed across all experimental settings. The figures are intended to illustrate this robust trend, which holds irrespective of specific configurations.

To ensure a fair and accurate comparison of computational cost in Figure 6, we measured the wall-clock time for the core training loop, explicitly excluding overhead from saving and loading model checkpoints. Each run was repeated multiple times to verify the stability of timing measurements. System resources were monitored to ensure that no extraneous processes interfered, and we confirmed that GPU thermal throttling did not impact the reported durations.

### I.1.3   Ablation Studies

The configurations required to replicate our ablation studies are detailed in Appendix F. The results can be directly obtained by executing the corresponding scripts in our codebase.

## I.2   Fair Baseline Comparisons

A central finding of this work is that existing methods often perform poorly on previously untested datasets, such as PathMNIST and TissueMNIST (Yang et al., 2023). This issue is not unique to this field; other areas of machine learning have faced similar challenges due to over-reliance on limited benchmark sets. Evaluation on a narrow range of datasets can create an illusion of progress and lead to indirect overfitting on test data through repeated hyperparameter tuning, even with proper validation splits (Hardt, 2025).

To mitigate this risk and promote generalizability, we evaluated all methods on previously untested datasets and architectures. Nevertheless, we ensured that baseline methods were given a fair chance by diligently tuning their hyperparameters. We adhered to the guidelines provided by the original authors, using their exact hyperparameters for established datasets. For new datasets, we started from the authors' recommended parameters and made reasoned adjustments to improve performance.

The time invested in hyperparameter search for many baseline methods was equal to or greater than that spent on our own model. While our search was necessarily limited by computational resources and the desire to avoid excessive fine-tuning, we carefully reasoned about the impact of key hyperparameters to achieve competitive results. Certain hyperparameter adjustments were found to benefit all methods uniformly on specific datasets; these improved, universally beneficial settings are the ones we report.

A key motivation for fast adversarial training is efficiency and practical viability (Wong et al., 2020). If extensive tuning is required to achieve competitive results, one might as well revert to more reli-

---

[1] https://anonymous.4open.science/r/2026_ICLR_SORA

able but computationally expensive methods like PGD. This rationale underpins our commitment to using a fixed hyperparameter set across all datasets and architectures for our method, while permitting necessary adjustments for baselines to ensure fairness.

Our hyperparameter search was intentionally constrained to PreActResNet-18 and WideResNet-28-10 on CIFAR-10 and CIFAR-100, since nearly all other baselines have also examined these settings (some use WideResNet-34-10 instead). Since these settings were not challenging enough, for our method and the baselines, we did some limited experiments on PATHMNIST as well. The resulting parameters were then fixed for evaluations on TISSUEMNIST and TINYIMAGENET (Le & Yang, 2015). For TINYIMAGENET, we incorporated published baseline hyperparameters where available; for TISSUEMNIST, no additional tuning was performed.

Notably, our method's key parameters ($\beta$ for reducing PertAlign variance, and $\alpha_0/\alpha_{\max}$ for step size constraints) required minimal tuning. All results on TISSUEMNIST and TINYIMAGENET represent **first-run outcomes without modification**, whereas baselines benefited from previously optimized settings when available.

## I.3 DATASET SELECTION CRITERIA

The selection of PATHMNIST and TISSUEMNIST for our evaluation was guided by several principled considerations. First and foremost, we sought datasets with comparable scale to established benchmarks like CIFAR-10 and CIFAR-100, both in terms of sample size and image dimensions (Table 2). This ensures that computational requirements remain manageable while maintaining relevance to real-world applications.

Medical imaging datasets were prioritized for two key reasons:

1. Robustness in medical applications carries significant practical importance, where model reliability can directly impact diagnostic outcomes, and

2. the distribution shift between natural images (e.g. CIFAR, IMAGENET) and medical images provides a rigorous test of generalization capabilities.

The MEDMNIST suite (Yang et al., 2023) emerged as a natural candidate due to its standardized formatting and accessibility.

Our selection process was systematic rather than exhaustive. We reviewed available datasets in the MEDMNIST collection and selected PATHMNIST and TISSUEMNIST based on their *number of training samples*, *test samples*, and their *number of classes*. We explicitly did not perform extensive dataset screening or cherry-picking to favor our method; these datasets were chosen from a limited candidate pool based on the above criteria within our available time constraints.

## I.4 LLM USAGE

Large Language Models (LLMs) were used to assist with proofreading and polishing the text of this manuscript. LLMs were also used for initial code scaffolding in some instances (primarily for generating plots or loading raw data). All code generated with LLM assistance was rigorously reviewed, tested, and validated by multiple co-authors to ensure its correctness and integration into our codebase.

