# OpenReview forum: "SORA: Free Second Order Attacks in Fast Adversarial Training"
_ICLR.cc/2026/Conference — Submitted to ICLR 2026_

### Official Review · Reviewer_PMUw · 2025-10-21

**Soundness:** 2
**Presentation:** 3
**Contribution:** 2
**Rating:** 4
**Confidence:** 4

**Summary:**

The authors propose an efficient metric to measure the non-linearity of loss landscape during adversarial training. By utilizing this metric, the optimal step size for adversarial attack can be calculated. The experiments on diverse datasets and architectures demonstrate the effectiveness of the proposed method.

**Strengths:**

1. The paper is written well
2. The method is computationally efficient
3. The theoretical analysis on PerAlign, which is to measure the non-linearity of the loss landscape, is rigorous

**Weaknesses:**

1. **Marginal improvement:** On CIFAR10 and CIFAR100, SORA only has slight improvement compared to NFGSM and AAER. What are the advantages of your method compared to them?
2. **Abnormal baseline performance:** The performance of GradAlign, ZeroGrad, ATAS, AAER is surprisingly low on PathMNIST and TissueMNIST. It is well known that AT is sensitive to hyperparameters. Did you tune their hyperparameters to ensure the optimal performance on these datasets?
3. **Some baselines are missing:** The results of critical baselines, e.g., ATTA [1], Fast-BAT [2], NuAT [3], are missing
4. **Lack of results on high-resolution datasets:** You only have the results on low-resolution datasets. It is still unknown whether your method can be scaled up to high-resolution datasets, e.g., ImageNet-100.

[1] Haizhong Zheng et al. Efficient adversarial training with transferable adversarial examples

[2] Yihua Zhang et al. Revisiting and advancing fast adversarial training through the lens of bi-level optimization.

[3] Gaurang Sriramanan et al. Towards efficient and effective adversarial training.

**Questions:**

See weaknesses

---

> ### Author Response · Authors · 2025-11-20
>
> We thank the reviewer for their detailed and thoughtful feedback. Below we address each concern.
>
> # W1: Concerns about marginal improvements
>
> We emphasize that **SORA is designed for generalization across diverse datasets and architectures (Section 1, Lines 53–77)**, specifically avoiding heavy dataset-specific or model-specific hyperparameter tuning **(Section 1, Lines 84–89; Appendix I.2, Lines 2211–2217)**. To ensure fair comparison, we even allowed dataset-specific tuning for the baselines on CIFAR-10/100 using hyperparameters recommended by the respective authors **(Appendix I.2, Lines 2200–2204)**, effectively strengthening their performance.
>
> Because these baselines are tuned specifically for these datasets, the remaining small gap on CIFAR-10/100 is expected. The true value of SORA becomes apparent on new or challenging datasets such as PathMNIST and TissueMNIST, where SORA achieves a clear and consistent advantage across all architectures. This improvement arises from SORA’s adaptive nature: unlike static one-step methods (e.g., NFGSM, AAER), SORA dynamically adjusts its step size based on the local loss landscape, enabling better clean-robust accuracy trade-offs without per-dataset and per-architecture calibration.
>
> Thus, while CIFAR datasets, where baselines are heavily optimized, show marginal gains, SORA’s key advantages emerge in generalization, robustness to dataset shift, and stability across architectures.
>
> # W2: Concerns about poor baseline performance
>
> Dependence on hyperparameter tuning for fast adversarial training methods is a critical flaw. Although many AT methods require hyperparameter tuning for different setting, we argue that for fast adversarial training methods, this dependence significantly diminishes their practicality, as we discuss in **Section 1, Lines 83–89**. **Performing costly hyperparameter searches for every dataset/model combination undermines the very goal of fast adversarial training**. In such a scenario, practitioners would reasonably default to PGD AT (e.g. PGD-5), which does not require delicate tuning and provides reliably strong robustness.
>
> For this reason, and to evaluate generalizability, we used author-recommended hyperparameters for all baselines **(Appendix B; Appendix I.2 Lines 2200–2210)**. Under this fair-comparison setting, methods such as GradAlign, ATAS, ZeroGrad, and AAER indeed perform well on CIFAR but fail to generalize to more challenging datasets like PathMNIST and TissueMNIST.
>
> These datasets exhibit:
>
> - Significant class imbalance **(Figure 7)**
>
> - Fine-grained textures and different statistical structure **(Figure 8)**
>
> Static or brittle methods tend to break under such conditions. SORA’s strong performance across all architectures and these harder datasets reflects its robust generalization and stability.
>
> # W4: ImageNet100 results
>
> We agree that experiments on higher-resolution datasets further strengthen our evaluation. As requested, we conducted additional experiments on ImageNet100. Due to computational constraints and the long training time on high-resolution data, we compared against competitive baselines with similar training cost; we will include additional baselines in the camera-ready version.
>
> The results below report clean, PGD-10, and AutoAttack (AA) accuracy for models trained on ImageNet100 using PreActResNet18 for 30 epochs with $\epsilon = 8/255$. SORA achieves the highest clean and robust accuracy among the compared methods.
>
> | Method   | Clean (%) | PGD-10 (%) | AA (%) |
> |----------|-----------|------------|--------|
> | **FGSM**  | 15.94     | 0.00       | 0.00   |
> | **NFGSM** | 49.40     | 21.44      | 15.52  |
> | **AAER**  | 48.26     | 23.38      | 17.18  |
> | **SORA**  | **57.26**     | **24.34**      | **18.56**  |

---

> > ### Comment · Reviewer_PMUw · 2025-11-24
> >
> > Thanks for your response, but my second concern still remains. I admit that a comprehensive hyper-parameter search undermines the goal of fast adversarial training. However, in this case, the experiment results can't convince me that your method outperform baselines by algorithm design rather than by better hyperparameter selection.
> >
> > How do you select the hyperparameters of your method? Are the hyperparameters just tuned based on CIFA10/CIFAR100 like the baselines did? I didn't see the ablation study on alpha and beta in Algorithm 1.

---

> > > ### Author Response · Authors · 2025-11-25
> > > **Comment 1/2: Response to Reviewer PMUw**
> > >
> > > We appreciate the reviewer’s commitment to algorithmic fairness. We fully agree that superior performance must be attributed to the method's design, not differential hyperparameter tuning. The detailed results you requested allow us to demonstrate this distinction comprehensively.
> > >
> > > # 1. SORA Calibration and Stability
> > >
> > > As noted in our manuscript **(Appendix I.2)**, our initial attempts to tune SORA on CIFAR-10/100 revealed that these datasets lack sufficient non-linearity to trigger our adaptive mechanism. In these smooth landscapes, SORA consistently defaulted to the maximum step size, preventing meaningful calibration of the sensitivity parameters ($\alpha$ and $\beta$).
> > > To ensure our mechanism was functional, we performed a minimal calibration on PathMNIST. Crucially, **this single set of parameters was then fixed and used universally for all other experiments** (TissueMNIST, TinyImageNet, ImageNet-100).
> > > To demonstrate that our method is robust (and not delicately "over-tuned"), we present the hyperparameter sweep for SORA on PathMNIST (PreActResNet-18) below.
> > >
> > > ## $α_{\max} = 1.5 \epsilon$
> > >
> > > | $\beta$  | ($\alpha_0=0.075$) Clean / PGD-10 | ($\alpha_0=0.1$) Clean / PGD-10 | ($\alpha_0=0.2$) Clean / PGD-10 |
> > > |-------|----------------------------|--------------------------|--------------------------|
> > > | 0.005 | 85.19 / 43.70              | 85.19 / 43.98            | 85.89 / 43.30            |
> > > | 0.01  | 85.79 / 42.77              | 86.28 / 40.19            | 85.19 / 45.43            |
> > > | 0.02  | 86.29 / 38.35              | 84.88 / 44.51            | 85.19 / 44.83            |
> > >
> > > ---
> > >
> > > ## $α_{\max} = 2.0 \epsilon$
> > >
> > > | $\beta$  | ($\alpha_0=0.075$) Clean / PGD-10 | ($\alpha_0=0.1$) Clean / PGD-10 | ($\alpha_0=0.2$) Clean / PGD-10 |
> > > |-------|----------------------------|--------------------------|--------------------------|
> > > | 0.005 | 86.54 / 41.68              | 83.20 / 46.44            | 84.67 / 28.24            |
> > > | 0.01  | 86.54 / 39.63              | 86.45 / 36.28            | 84.41 / 30.54            |
> > > | 0.02  | 86.74 / 39.88              | 86.43 / 40.71            | 67.78 / 06.43            |
> > >
> > > ---
> > >
> > > ## $α_{\max} = 2.5 \epsilon$
> > >
> > > | $\beta$  | ($\alpha_0=0.075$) Clean / PGD-10 | ($\alpha_0=0.1$) Clean / PGD-10 | ($\alpha_0=0.2$) Clean / PGD-10 |
> > > |-------|----------------------------|--------------------------|--------------------------|
> > > | 0.005 | 75.12 / 32.96              | 81.69 / 41.72            | 70.19 / 06.97            |
> > > | 0.01  | 80.76 / 22.49              | 65.72 / 18.77            | 87.20 / 18.98            |
> > > | 0.02  | 67.50 / 29.01              | 25.36 / 29.61            | 75.77 / 28.69            |

---

> > > ### Author Response · Authors · 2025-11-25
> > > **Comment 2/2: Response to Reviewer PMUw (continued)**
> > >
> > > # 2. Baseline Brittleness and Lack of Generalization
> > >
> > > To directly address your concern that baselines failed simply due to a lack of tuning, we conducted the requested comprehensive hyperparameter search for NFGSM, ELLE, GradAlign, AAER, and ZeroGrad on PathMNIST. Since ATAS is not competitive even on CIFAR10/100 datasets we instead did hyperparameter sweep for NFGSM and ELLE.
> > >
> > > The results confirm that even with dedicated tuning, **these methods generally fail to find a stable configuration. Furthermore, for the few configurations that do work on PathMNIST, performance drops significantly when those same parameters are applied back to CIFAR-10**. This proves these methods lack a single, generalizable hyperparameter set.
> > >
> > > **NFGSM** Sweep (PathMNIST vs. CIFAR-10 Check)
> > > Note: The best PathMNIST setting ($K=2\epsilon$, $\alpha=0.5\epsilon$) causes a ~9% robust accuracy drop on CIFAR-10 compared to standard settings.
> > >
> > > | $K$                  | $\alpha$                 | Dataset      | Clean (%) | PGD-10 (%) |
> > > |--------------------|-------------------|-----------------------|-----------|------------|
> > > | $\epsilon$                  | $\epsilon$                 | PathMNIST             | 86.04     | 0.82       |
> > > | $3\epsilon$                 | $\epsilon$                 | PathMNIST             | 62.71     | 20.59      |
> > > | $3\epsilon$                 | $2\epsilon$                | PathMNIST             | 66.78     | 4.45       |
> > > | $2\epsilon$                 | $2\epsilon$                | PathMNIST             | 35.94     | 1.15       |
> > > | $2\epsilon$                 | $0.75\epsilon$             | PathMNIST             | 59.42     | 27.61      |
> > > | $2\epsilon$                 | $0.5\epsilon$              | PathMNIST             | 85.16     | 37.08      |
> > > | $2\epsilon$                 | $0.5\epsilon$              | CIFAR10 	| 86.57     | 39.84      |
> > >
> > > **ELLE** Sweep (PathMNIST vs. CIFAR-10 Check) Note: Higher regularization ($\lambda$) improves PathMNIST but degrades CIFAR-10 performance.
> > >
> > > | $\lambda$| Dataset    | Clean (%) | PGD-10 (%) |
> > > |--------|------------|-----------|------------|
> > > | 4000   | PathMNIST  | 83.52     | 43.08      |
> > > | 8000   | PathMNIST  | 79.55     | 42.77      |
> > > | 12000  | PathMNIST  | 78.82     | 42.87      |
> > > | 16000  | PathMNIST  | 77.70     | 44.21      |
> > > | 20000  | PathMNIST  | 70.04     | 38.82      |
> > > | 4000   | CIFAR10    | 81.12     | 44.27      |
> > > | 8000   | CIFAR10    | 79.10     | 42.63      |
> > > | 12000  | CIFAR10    | 76.55     | 41.35      |
> > > | 16000  | CIFAR10    | 74.90     | 40.70      |
> > > | 20000  | CIFAR10    | 73.52     | 39.52      |
> > >
> > > **GradAlign, AAER, & ZeroGrad (Persistent Failure)** Even with extensive sweeps, these methods either suffered from Catastrophic Overfitting (CO) or achieved very low robustness on PathMNIST.
> > >
> > > GradAlign:
> > >
> > > | $\lambda$ | Dataset    | Clean (%) | PGD-10 (%) |
> > > |--------|------------|-----------|------------|
> > > | 0.4    | PathMNIST  | 11.39     | 0.19       |
> > > | 0.6    | PathMNIST  | 9.87      | 0.54       |
> > > | 0.8    | PathMNIST  | 73.26     | 8.65       |
> > > | 1.0    | PathMNIST  | 35.58     | 1.81       |
> > >
> > >
> > > AAER:
> > >
> > > | $\lambda_1$ | $\lambda_2$ | $\lambda_3$ | Dataset    | Clean (%) | PGD-10 (%) |
> > > |---------|---------|---------|------------|-----------|------------|
> > > | 1       | 5       | 0.55    | PathMNIST  | 82.26     | 32.58      |
> > > | 1       | 8.5     | 0.55    | PathMNIST  | 82.26     | 32.58      |
> > > | 1       | 2.75    | 0.55    | PathMNIST  | 82.26     | 32.58      |
> > > | 1       | 5       | 1.5     | PathMNIST  | 18.64     | 18.64      |
> > > | 1       | 8.5     | 1.5     | PathMNIST  | 18.64     | 18.64      |
> > > | 1       | 2.75    | 1.5     | PathMNIST  | 18.64     | 18.64      |
> > > | 1       | 5       | 0.75    | PathMNIST  | 18.64     | 18.64      |
> > > | 1       | 8.5     | 0.75    | PathMNIST  | 18.64     | 18.64      |
> > > | 1       | 2.75    | 0.75    | PathMNIST  | 18.64     | 18.64      |
> > >
> > > ZeroGrad:
> > >
> > > | $q_{val}$ | Dataset    | Clean (%) | PGD-10 (%) |
> > > |-------|------------|-----------|------------|
> > > | 0.35  | PathMNIST  | 49.47     | 0.86       |
> > > | 0.60  | PathMNIST  | 78.69     | 8.18       |
> > > | 0.70  | PathMNIST  | 88.13     | 10.43      |
> > > | 0.90  | PathMNIST  | 88.32     | 16.42      |

---

> > > > ### Comment · Reviewer_PMUw · 2025-11-26
> > > >
> > > > I appreciate the supplementary results. My concerns have been addressed. I will raise the score to 6.

---

> > > > > ### Author Response · Authors · 2025-11-27
> > > > >
> > > > > Thank you once again for your valuable comments, which have helped us improve the clarity and rigour of our manuscript. Please know that we look forward to addressing any further concerns or questions regarding our work.

---

> ### Author Response · Authors · 2025-11-27
>
> Following up on your suggestion regarding missing baselines, we prioritized running NuAT during the rebuttal period to provide a preliminary comparison for the official record. While we still commit to adding ATTA and Fast-BAT in the camera-ready version, these immediate results for NuAT further reinforce our discussion on algorithmic stability.
>
> ### Methodology:
> We adhered to our fair comparison protocol: we used the authors’ recommended $\lambda$ settings where available. For datasets/architectures not specified in the original paper (e.g., CIFAR100, TinyImageNet, PathMNIST, TissueMNIST), we tested the $\lambda$ values reported for CIFAR-10/ImageNet-100 ($\lambda \in \{1.0, 4.0\}$) to probe generalization.
>
> ### Summary of Results:
> The results (averaged over 3 seeds) reinforce our paper's core finding: **NuAT exhibits significant brittleness and instability outside of its original tuning scope.** It frequently suffers from Catastrophic Overfitting (CO) or severe performance drops on PathMNIST, TissueMNIST, and even specific architectures on CIFAR-10. Furthermore, **NuAT incurs higher computational overhead due to extra forward/backward passes compared to SORA.**
>
> ### CIFAR-10 ($\lambda = 4.0$)
>
> | Architecture     | Clean (%)         | FGSM (%)          | PGD-10 (%)        |
> |------------------|------------------:|------------------:|------------------:|
> | PreActResNet18   | 75.09 ± 0.19      | 53.10 ± 0.17      | 50.28 ± 0.21      |
> | ResNet18         | 84.32 ± 6.57      | 53.88 ± 1.26      | 35.53 ± 10.16     |
> | WideResNet28     | 86.53 ± 1.91      | 56.08 ± 5.33      | 29.37 ± 2.82      |
> | SENet18          | 86.39 ± 2.56      | 56.69 ± 5.08      | 30.74 ± 5.14      |
>
> ### CIFAR-100 ($\lambda = 1.0$)
>
> | Architecture     | Clean (%)         | FGSM (%)          | PGD-10 (%)        |
> |------------------|------------------:|------------------:|------------------:|
> | PreActResNet18   | 49.66 ± 0.41      | 29.23 ± 0.29      | 27.81 ± 0.33      |
> | ResNet18         | 49.28 ± 0.43      | 29.28 ± 0.35      | 27.84 ± 0.28      |
> | WideResNet28     | 30.98 ± 29.98     | 22.85 ± 21.85     | 11.07 ± 10.07     |
> | SENet18          | 47.58 ± 0.24      | 28.41 ± 0.43      | 26.96 ± 0.25      |
>
> ### PathMNIST ($\lambda = 4.0$)
>
> | Architecture     | Clean (%)         | FGSM (%)          | PGD-10 (%)        |
> |------------------|------------------:|------------------:|------------------:|
> | PreActResNet18   | 82.73 ± 2.96      | 66.51 ± 4.54      | 25.69 ± 3.93      |
> | ResNet18         | 88.45 ± 1.62      | 53.96 ± 1.64      | 17.73 ± 2.60      |
> | WideResNet28     | 87.79 ± 1.85      | 55.68 ± 9.04      | 21.46 ± 1.23      |
> | SENet18          | 86.37 ± 1.12      | 55.33 ± 7.60      | 24.19 ± 1.34      |
>
> ### TissueMNIST ($\lambda = 4.0$) (PreActResNet18, Seed 42)
>
> | Metric          | Value (%) |
> |-----------------|----------:|
> | Clean           | 66.33     |
> | FGSM            | 29.42     |
> | PGD-10          | 6.38      |
>
> ### CIFAR100 ($\lambda = 4.0$)
>
> | Seed | Clean (%) | FGSM (%) | PGD-10 (%) |
> |------|-----------:|----------:|------------:|
> | 42   | 6.90       | 6.48      | 6.44        |
> | 33   | 6.79       | 6.39      | 6.37        |
> | 21   | 6.62       | 6.31      | 6.22        |
>
> ### TinyImageNet ($\lambda = 4.0$)
>
> | Seed | Clean (%) | FGSM (%) | PGD-10 (%) |
> |------|-----------:|----------:|------------:|
> | 42   | 0.50       | 0.50      | 0.50        |
> | 33   | 0.50       | 0.50      | 0.50        |
> | 21   | 0.50       | 0.50      | 0.50        |
>
> ### TinyImageNet ($\lambda = 1.0$)
>
> | Seed | Clean (%) | FGSM (%) | PGD-10 (%) |
> |------|-----------:|----------:|------------:|
> | 42   | 32.29      | 19.71     | 18.96       |
> | 33   | 31.79      | 20.08     | 19.27       |
> | 21   | 40.32      | 34.36     | 10.44       |

---

### Official Review · Reviewer_pwii · 2025-10-28

**Soundness:** 2
**Presentation:** 3
**Contribution:** 2
**Rating:** 4
**Confidence:** 3

**Summary:**

This paper addresses the problem of Catastrophic Overfitting (CO) in single-step adversarial training (AT). The authors identify a phenomenon termed "Epsilon Overfitting" (EO), where models overfit to the specific perturbation magnitude used during training. They propose Perturbation Alignment (PertAlign) to predict the onset of CO, and leverage these insights to develop SORA, an adaptive step-size AT method. SORA dynamically adjusts perturbation sizes based on an approximation of the local loss curvature. Extensive experiments across multiple datasets and architectures show that SORA can prevent CO and achieve competitive robust accuracy with minimal computational overhead.

**Strengths:**

1.	Comprehensive Evaluation: The paper provides a thorough empirical evaluation across a diverse set of datasets (including challenging medical imaging benchmarks) and model architectures.
2.	Identification of Epsilon Overfitting (EO): The observation that fixed, large perturbation magnitudes can lead to overfitting on specific ε values is an interesting and well-documented analysis.
3.	Low-Cost Metric: The proposed PertAlign metric is computationally cheap, as it reuses gradients already computed during the standard training process.
4.	Practical Algorithm: The SORA algorithm itself is relatively simple to implement and integrates seamlessly into existing fast AT pipelines.

**Weaknesses:**

1.	Significance of the Core Problem: A fundamental question remains: How critical is Catastrophic Overfitting in the broader landscape of adversarial robustness? While CO is a known failure mode in single-step AT, the paper does not sufficiently articulate why CO remains a pressing, unsolved issue that warrants a new solution, especially given that multi-step methods like PGD-10 are still considered the gold standard for high robustness, albeit at a higher cost.
2.	Novelty and Depth of Contributions: 1) Epsilon Overfitting (EO): While the term is new, the underlying idea—that fixed-step attacks can lead to non-robust, overfitted decision boundaries—is a core intuition behind many existing adaptive and multi-step methods. The claim that EO is a "previously overlooked phenomenon" may be an overstatement; it is more accurately a new and specific characterization of a known class of problems. 2) Perturbation Alignment (PertAlign): The theoretical derivation connects PertAlign to the Hessian, which is a solid contribution. However, from a practical standpoint, PertAlign can be perceived as a relatively trivial trick: it is essentially the cosine similarity between the gradient at a random start and the gradient after one FGSM step. What is it essentially different from GradAlign?
3.	Marginal Practical Gains: The experimental results show that SORA's advantages are often marginal. In many tables, SORA's robust accuracy is within less than ~1% of the best-performing single-step baselines (e.g., NFGSM, AAER). While it achieves better clean accuracy on some datasets like PathMNIST, the overall improvement in the trade-off between robustness and accuracy is not dramatic. Hence, the net practical benefit of adopting SORA is not overwhelmingly compelling.
4.	Theoretical Grounding Justification: The theoretical analysis in Appendix A is technically sound, deriving the optimal step size under a quadratic loss assumption. However, the justification for this approximation in the context of deep neural networks is weak. The highly non-convex and complex loss landscapes of modern DNNs are far from quadratic, and the paper does not provide evidence that this local approximation holds well enough in practice to be truly meaningful, beyond serving as a heuristic inspiration for the algorithm.
5.	Comparison to Competitors: The paper positions SORA as a state-of-the-art method. However, when compared to strong baselines like PGD-10 or TRADES, SORA consistently lags in robust accuracy (as expected, given their higher computational cost). Among its single-step peers, it is a strong contender but not a clear dominator. The claim of "superior efficiency" is true relative to multi-step methods, but its advantage in time/memory over other single-step methods (see Figure 6) is minimal, and its performance gain is similarly slight.

**Questions:**

See the weaknesses.

1.	Beyond preventing CO, what specific, significant advantage does SORA offer over other CO-robust single-step methods, given that the accuracy improvements are often marginal?
2.	Can you provide empirical evidence validating the quadratic loss assumption in deep networks? How sensitive is SORA's performance to scenarios where this assumption breaks down?
3.	The concept of adapting to local curvature is not new. How is SORA's approach fundamentally different or better than prior adaptive step-size methods like ATAS, beyond the specific heuristic used?

---

> ### Author Response · Authors · 2025-11-15
> **Comment 1/2: Response to Reviewer pwii**
>
> We appreciate the reviewer’s comments and the chance to clarify our contributions. Below, we address each point in detail, referencing the relevant parts of the manuscript.
>
> # W1: Significance of the Core Problem
>
> CO remains a pressing and unsolved challenge. Two main issues persist: (i) **Practically**, CO is the most prominent failure point of single-step AT methods. While many works perform well on CIFAR-10/100 **(Section 1, Lines 50-90; Section 6.3)**, they lack generalizability across datasets and architectures or exhibit poor clean-robust accuracy trade-offs. (ii) **Theoretically**, CO as a phenomenon is still not completely understood, particularly regarding its root cause and why models undergo this sudden transition. The continued publication of CO-focused works at venues such as ICLR [1] (2024) and NeurIPS [2] (2023), [3] (2022) in recent years demonstrates its ongoing relevance. While PGD-10 is reliable, its significant computational cost limits scalability to larger datasets and architectures, and it consumes roughly 10× the energy of methods such as FGSM. necessitating efficient single-step alternatives.
>
> # W2: Novelty and Depth of Contributions
>
> **(1)** Previous studies have highlighted the limitations of fixed-budget or step-size attacks [6]. However, in the setting of single-step adversarial training, new phenomena arise. One such issue is Epsilon Overfitting, which could even result in FGSM accuracy exceeding the clean accuracy when evaluated with the same training epsilon. This reflects a tendency to overfit specifically to the chosen epsilon values **(Section 3 lines 194–199, Figure 9)**. Unlike prior work that primarily addresses multi-step attacks, our focus is on fast adversarial training, which introduce distinct challenges (e.g. CO).
>
> **(2) PertAlign's simplicity is a strength, not a weakness**, it provides an easy-to-use, reliable metric for CO prediction. As detailed in **Appendix C.1**, PertAlign differs fundamentally from GradAlign: (i) GradAlign emerges as a special case within our second-order framework **(Lemma 4.1)**, (ii) GradAlign is used as a regularizer with computational overhead, while PertAlign serves purely as a monitoring metric with near-zero cost, and (iii) **Figure 6** shows GradAlign incurs 2-3× higher computation and memory overhead, making it impractical for monitoring during training. **PertAlign enables real-time CO prediction for any single-step method without adding overhead** with examples shown in **Figure 11**.
>
> # W3 - Q1: Performance of SORA
>
> We respectfully clarify the performance gap between SORA and the baselines. It can be seen from **Figure 5 Left** that although baselines like NFGSM [3] and AAER [2] match and compete with SORA on datasets like CIFAR10, CIFAR100, and TinyImageNet, they fail on all the architectures tested on PathMNIST, demonstrating the lack of generalizability for these methods compared to our method. Also from **Figure 5 Right** it can be seen that even on those datasets that SORA’s robust accuracy is similar to NFGSM and AAER, SORA has higher clean accuracy, demonstrating SORA has better robust and clean accuracy trade off. On the other hand methods such as ELLE [1] and MultiGrad [4] which perform better than NFGSM and AAER, incur significant more computation and memory overhead as discussed in **Section 6.3 Lines 470-475** and shown in **Figure 6**, Meaning SORA is the only method which can reliably achieve or surpass state-of-the-art performance across all datasets and architectures while being very efficient.
>
> # W4 - Q2: Theoretical Grounding Justification
>
> Robust models tend to have a smoother loss landscape. We have visualized this in **Figure 1 Left** and support it empirically in **Figure 2 b** where the robust accuracy declines smoothly as a function of epsilon which demonstrates this semi linearity of loss surface. If the model is in CO state, although the loss surface becomes non-linear, as is shown in **Figure 2 a** and discussed in **Section 3 Lines 234-242**, this non-linearity is not jagged and has a characteristic to it, thus a quadratic model can be a good choice. The effectiveness of our method across various datasets and architectures provides strong empirical evidence that the local quadratic approximation holds sufficiently well in practice. When it doesn't, our random sampling mechanism **(Section 5, Lines 366-371)** compensates by increasing perturbation diversity.
>
> We also note that nearly all single-step fast AT methods rely on first-order approximations of the loss landscape, which are inherently less accurate than a second-order approximation.

---

> ### Author Response · Authors · 2025-11-15
> **Comment 2/2: Response to Reviewer pwii (continued)**
>
> # W5: Comparison to Competitors
>
> As the reviewer acknowledged, multi-step methods naturally achieve higher robustness by sacrificing efficiency, direct comparison is therefore inappropriate. Among single-step methods, all require at least one forward and backward pass for adversarial example generation, establishing a fundamental efficiency upper bound. SORA operates at this upper bound (matching FGSM-RS [5] and NFGSM [3] in cost) while achieving superior clean accuracy and cross-dataset generalization. This represents the optimal point in the efficiency-robustness trade-off space for single-step methods.
>
> # Q3: How is SORA fundamentally different from ATAS?
>
> **SORA uses principled second-order approximation, while ATAS relies on first-order heuristics**. Although both methods use adaptive step size, ATAS updates step size heuristically and solely depending on the norm of the gradient, which is a first order approximation, but SORA step size selection is theoretically justified and uses second order approximation which is more accurate.
>
> Moreover, ATAS requires longer training time and significantly higher memory consumption **(Figure 6)**, while also delivering worse performance **(Figure 5, Table 1)**.
>
> # References
>
>  [1] Efficient local linearity regularization to overcome catastrophic overfitting
>
>  [2] Eliminating Catastrophic Overfitting Via Abnormal Adversarial Examples Regularization
>
>  [3] Make Some Noise: Reliable and Efficient Single-Step Adversarial Training
>
>  [4] ZeroGrad: Mitigating and Explaining Catastrophic Overfitting in FGSM Adversarial Training
>
>  [5] Fast is better than free: Revisiting adversarial training
>
>  [6] MMA Training: Direct Input Space Margin Maximization through Adversarial Training

---

> ### Author Response · Authors · 2025-11-27
>
> Dear Reviewer pwii,
>
> Thank you again for your thoughtful review. Since the discussion period is short, we wanted to briefly highlight how our rebuttal directly addressed your main concerns.
>
> ### 1. EO and PertAlign introduce new, actionable insights.
> - EO is not just a fix step size effect. EO stems from low perturbation diversity which is not covered by prior work.
> - PertAlign is not GradAlign: it has a **second-order derivation**, **measures curvature along adversarial directions as opposed to random direction**, **zero-cost**, and is not used as regularizer, unlike GradAlign/ELLE.
> - This makes PertAlign predictive of CO, not reactive.
>
> ### 2. SORA’s benefit is stability and generalization, not just CIFAR improvements.
> **SORA is the only single-step method that stays stable across all datasets/architectures, including PathMNIST and ImageNet-100 (shown in the response of reviewer PMUw).** NFGSM/AAER fail on these, while ELLE/MultiGrad are much slower and still weaker.
>
> ### 3. Quadratic approximation is empirically supported.
> Figures 1–2 show smooth curvature patterns in both robust and CO regimes, and SORA works well even when the approximation is imperfect; our randomization step compensates when needed. Existing fast AT methods rely only on first-order approximations, ours is strictly stronger.
>
> ### 4. Difference from ATAS.
> ATAS uses a first-order gradient-norm heuristic. SORA uses a principled second-order step-size criterion, while training faster, using less memory, and achieving better accuracy.
>
> We would be happy to clarify any remaining questions. Your feedback is sincerely appreciated, and any reconsideration would mean a great deal to us.

---

### Official Review · Reviewer_Xi2k · 2025-10-28

**Soundness:** 3
**Presentation:** 3
**Contribution:** 2
**Rating:** 2
**Confidence:** 4

**Summary:**

This paper introduces and analyze Epsilon Overfitting, demonstrating its importance for robust generalization and CO. The authors propose PertAlign, a efficient metric for early and reliable CO prediction. The authors verify the effectiveness of their method on different methods and datasets.

**Strengths:**

1.	The motivation and method are clear, and the visualization is helpful.
2.	The authors provide sufficient evidence, including extensive experiments and ablation studies, to support the effectiveness of their proposed method.
3.	This paper is well-written, making it easy to follow.

**Weaknesses:**

1.	Misleading or Overstated Claims
(1) The fact that fixed perturbation magnitudes exacerbate CO has been thoroughly studied in prior work [1,2]. It is already well-known that larger perturbations easily lead to CO [3,4]. This is not a “previously overlooked phenomenon” as the authors claim.
(2) The paper argues that existing measures incur a high computational cost. But [4, 5] already provides a time-efficient CO indicator without extra backwards.
2.	Weak Novelty Foundation
(1)	The “epsilon overfitting” part merely repeats existing research on decision boundary distortion. Many works already explain and measure distorted boundaries. The more important insight is to explain: how does boundary distortion dynamically during training, and what causes the spontaneous and initial onset of distortion.
(2)	PertAlign is fundamentally the same as GradAlign, with only a different radius for landscape measurement. Moreover, various efficient landscape-measurement techniques already exist [5].
3.	Effectiveness of the Method
(1)	Most machine learning method requires hyperparameter searching per dataset and architecture. Using “universal hyperparameters” is beneficial but not an excuse to deny baseline performance.  The authors should ensure fair comparisons by tuning baselines on different dataset and architecture, such as SENet and PathMNIST.
(2)	The method reduces CO by dynamically decreasing step-size, which raises the concern that it simply converges to a trivial solution where the perturbation strength is too small to trigger CO. To rule out this trivial solution, the authors must verify performance under more challenging perturbation budgets (e.g., 32/255), since N-FGSM maintains robustness at 16/255.

[1] Understanding Catastrophic Overfitting in Single-step Adversarial Training
[2] Fast Adversarial Training with Adaptive Step Size
[3] Fast is better than free: Revisiting adversarial training
[4] Eliminating Catastrophic Overfitting Via Abnormal Adversarial Examples Regularization
[5] Efficient local linearity regularization to overcome catastrophic overfitting

**Questions:**

Refer to weaknesses.

---

> ### Author Response · Authors · 2025-11-15
>
> We thank the reviewer for their thoughtful feedback. We have addressed the points below, and we believe the concerns can be resolved by referring to specific sections, figures, and results already present in our manuscript.
>
> # W1 - Concerns about claims
> **(1)** We respectfully clarify that our contribution does not concern the effect of perturbation magnitude alone, which prior work has indeed studied. Works such as [1] analyze vulnerability under smaller step sizes during CO, while [2] adapt the step size based on gradient-norm growth. In contrast, our finding is that **limited variability in the direction and distribution of perturbations**, even when expected magnitude is fixed, directly increases the likelihood of CO. This mechanism is distinct from simply using a larger ε.
>
> This distinction also explains why randomizing perturbation channels/pixels **(Section 5, Lines 366–371; Appendix F.1, Table 3)** significantly mitigates CO compared to simple FGSM training, despite having the same average perturbation magnitude, a behavior not accounted for in prior work.
>
> Finally, we note that for sufficiently small ε, neither CO nor Epsilon Overfitting occurs, so our analysis is explicitly concerned with the standard ε ranges where fast adversarial training exhibits CO.
>
> **(2) Figure 3** demonstrates PertAlign is predictive, not reactive. AAEs [3] and ELLE [4] increase after FGSM-PGD accuracy divergence (i.e., after CO occurs), while PertAlign collapses before divergence, enabling proactive intervention. Additionally, PertAlign incurs near-zero overhead by reusing training gradients **(Section 4 Lines 294-299)**, while ELLE requires two additional forward passes with cost comparable to a backward pass.
>
> # W2 - Concerns about novelty
> **(1)** Epsilon Overfitting specifies that adversarial training with limited perturbation variability causes models to expand decision boundaries only around perturbation landing points, rather than linearizing the loss across the entire ℓ∞ ball. As the reviewer also mentioned regarding the importance of explaining the evolution of boundary distortion, we have visualized this in **Figures 1 and 4**, while **Figure 2** supports this empirically.
>
> **(2)** PertAlign extends beyond GradAlign. **Section C.1** details the key differences:
> (i) we establish a connection to the loss landscape Hessian via **Lemma 4.1**, which is absent in the original GradAlign paper;
> (ii) PertAlign requires no additional backward passes, providing a 2–3× speedup **(Figure 6)**;
> (iii) our formulation more effectively captures geometric properties that precede CO.
>
> While GradAlign appears as a special case within our framework, our contributions extend significantly beyond it. As reviewer **ypja** also noted, PertAlign can be used in a “plug-and-play manner.” Furthermore, GradAlign acts as a regularizer, which overconstrains the model and reduces clean accuracy, while PertAlign is used only to identify non-linearity and adaptively modify the attack, without restricting the model. Unlike GradAlign, which measures non-linearity in a random direction, PertAlign measures non-linearity along the perturbation (optimization) direction, indicating whether the chosen step size has exceeded the locally smooth region.
>
> # W3 - Concerns about SORA
> **(1)** Hyperparameter universality is critical for practical fast adversarial training; if extensive per-dataset tuning is required, practitioners would simply use standard multi-step AT. We ensured fair baseline comparisons by using authors' recommended hyperparameters **(Section B, Section I.2 Lines 2200-2210)**. We are happy to provide additional results if the reviewer has specific concerns.
>
> **(2)** SORA does not converge to trivially small perturbations. Our strong performance against PGD and AutoAttack **(Tables 7–24)** confirms robustness across the full ℓ∞ ball, if perturbations became too small, these attacks would succeed. This indicates that SORA’s perturbation magnitudes remain substantial throughout training. **Table 5** demonstrates robust performance when trained with different ε values, and **Table 6** demonstrates robust evaluation for a model trained with ε=8/255 and tested across varying ε values.
>
> Regarding ε=32/255: single-step methods generally perform poorly at such large ε values (e.g., Table 2 of [4] shows near-random performance at ε=26/255 on CIFAR), as expected given the trade-off between efficiency and accuracy. Moreover, such perturbation sizes violate perceptual imperceptibility [6], a core requirement for adversarial examples.
>
> # References
>
> [1] Understanding Catastrophic Overfitting in Single-step Adversarial Training
>
> [2] Fast Adversarial Training with Adaptive Step Size
>
> [3] Eliminating Catastrophic Overfitting Via Abnormal Adversarial Examples Regularization
>
> [4] Efficient local linearity regularization to overcome catastrophic overfitting
>
> [5] Understanding and Improving Fast Adversarial Training
>
> [6] Intriguing properties of neural networks

---

> ### Author Response · Authors · 2025-11-27
>
> Dear Reviewer Xi2k,
>
> Thank you again for your time and thoughtful feedback. The main concerns expressed are (a) PertAlign vs. GradAlign, (b) the novelty of Epsilon Overfitting, (c) SORA’s effectiveness, and (d) fairness in baseline evaluation. For convenience, we summarize the key clarifications from our rebuttal:
>
> ### 1. PertAlign is fundamentally different from GradAlign.
> - PertAlign is derived from a second-order analysis and measures curvature along the actual adversarial update direction, unlike GradAlign’s random-direction alignment.
>  - It incurs **zero additional cost** because it reuses existing gradients; GradAlign/ELLE require **extra forward/backward passes**.
>  - PertAlign is **predictive of CO**, whereas ELLE/AAE only **react after CO** occurs.
>
> ### 2. Epsilon Overfitting is not a large-$\epsilon$ effect.
> EO stems from low perturbation diversity, not perturbation size. This explains why pixel/channel randomization with the same average $\epsilon $ strongly reduces CO, a behavior not captured by prior work.
>
> ### 3. SORA does not shrink to trivial perturbations.
> PGD/AA evaluations and $\epsilon$-sensitivity tests show that SORA maintains substantial perturbation strength throughout training.
>
> ### 4. Fair baseline comparison.
> We followed author-recommended hyperparameters for all baselines (as stated in the manuscript). Additionally, in our response to reviewer PMUw, **we conducted full hyperparameter sweeps on PathMNIST for multiple baselines, showing that they remain brittle even when tuned**, supporting our claims about generalizability.
>
> We understand the discussion period is short and would be happy to clarify anything further. Your response would be greatly appreciated.

---

### Official Review · Reviewer_ypja · 2025-11-06

**Soundness:** 3
**Presentation:** 3
**Contribution:** 2
**Rating:** 4
**Confidence:** 5

**Summary:**

This paper investigates the catastrophic overfitting issue in accelerated adversarial training and propose a metric called PertAlign to predict  CO in advance. PertAlign is based on the second order analyses on the input loss landscape and can be used to adaptive choose the step size when applying FGSM in adversarial training. The authors provide both theoretical motivations and the experimental results to validate the method (SORA) proposed.

**Strengths:**

++ PertAlign is theoretically motivated and is a low-cost indicator predicting CO in advance.

++ The method is simple and in a plug-and-play manner.

++ The experiments are relatively comprehensive on various models and datasets. The time / memory scatter shows SORA in competitive among fast adversarial training methods.

**Weaknesses:**

I have the following concerns about this paper:

1. Limited novelty: the relationship between the craggy loss landscape and catastrophic overfitting is actually not new.

2. The motivation of PertAlign is based on the second order approximation (Lemma 4.1). However, the step size $\alpha$ is generally quite large in one-step adversarial training, sometimes it can even be larger than $\epsilon$, I am not sure if the higher-order terms in the proof of Lemma 4.1 can be ignored. If not ignored then the approximation will not be accurate.

3. While the experiments are comprehensive, the performance improvement on AA, the most reliable evaluation, is very marginal (From Table 7 to 24). In most tables, the performance difference between SORA and the strongest baseline among fast adversarial training methods, is very small and smaller than the performance variance.

4. The experiments focus on the $l_\infty$ attacks and image classification problems, it would be better to include $l_2$ or $l_1$ perturbations and other tasks. In addition, the SORA pseudo-code seems to consider $l_\infty$ perturbations only, is it applicable to $l_2$ or $l_1$ cases? If not, what modifications do we need to make?

5. In addition to Table 3, more ablation studies are expected, such as sweeping the values of new hyper-parameters ($\alpha_max$, $\alpha_0$, $\beta$ in SORA's pseudo-code)

Minor issues:

1. Some missing literature:

    * About second-order curvature in adversarial training: "Robustness via Curvature Regularization" (CVPR 2019)
    * About fast adversarial training or geometry: "YOPO: You only propagate once" (NeurIPS 19),

2. In Table 5's caption: "The values in each cell correspond to clean, FGSM and PGD-10 accuracies." while I see only two results per cell.

**Questions:**

Please address the questions in the weakness section:

1. What is the key difference between PertAlign and existing work? (incl. the missing literature mentioned) If using second-order approximation is the key, then it would be better to demonstrate how tight the second order approximation is in the input loss landscape.

2. Why the improvement is marginal? What is the practical advantages of the proposed method?

3. The method's performance on tasks other than image classification.

4. The adaptation of the proposed methods to $l_2$ and $l_1$ cases and the corresponding experimental results.

5. More ablation studies on the newly introduced hyper-parameters.

---

> ### Author Response · Authors · 2025-11-20
> **Comment 1/2: Response to Reviewer ypja**
>
> We thank the reviewer for the detailed and thoughtful feedback. Below we address each concern and report additional results where requested.
>
> # W1: Limited novelty (loss cragginess and CO)
>
> We agree that prior work has connected non-linear (craggy) loss landscapes with Catastrophic Overfitting (CO). As we state in **Section 3 Lines 180 – 182**, CO is accompanied by increased non-linearity. Our contribution is a characterization of this non-linearity’s shape and dynamics: we show (visually in **Figure 1** and empirically in **Figure 2**) that the non-linearity that precedes CO is not arbitrarily jagged but follows a relatively smooth, structured curve. This observation directly motivates PertAlign and SORA **Section 3, Lines 234–242**, enabling a low-cost predictor and an effective adaptive step-size rule. We view EO as a specific characterization of the broader phenomenon, one that yields a practical diagnostic (PertAlign) and a simple, effective remedy (SORA).
>
> # W2: Validity of the second-order approximation (Lemma 4.1)
>
> We acknowledge the reviewer’s concern that the single-step sizes used in fast AT may be large and that higher-order terms could matter. Several points address this concern:
>
> - First, perceptual imperceptibility constrains the perturbation norm to be significantly smaller than the image norm (see e.g., [1]), which bounds the regime of interest.
>
> - Second, while any local quadratic approximation is an approximation, our empirical results **(Figure 11)** show that PertAlign, derived from the second-order view, reliably predicts CO across architectures and datasets.
>
> - Third, when the quadratic approximation is less accurate, our randomization mechanism **(Section 5, Lines 366–371)** increases perturbation diversity, which empirically compensates for approximation errors.
>
> Thus, although the approximation is not exact everywhere, it is sufficiently accurate in practice for CO prediction and for guiding SORA’s adaptive step-size decisions.
>
> # W3 - Q2: Concerns about performance of SORA
>
> We respectfully clarify that, aside from SORA, none of the competitive baselines maintain consistent performance across different datasets and architectures **(Figure 5)**. For example, while NFGSM and AAER achieve competitive robust accuracy on CIFAR-10/100 and TinyImageNet, their performance drops substantially across all tested architectures on PathMNIST. Conversely, methods such as ELLE and MultiGrad, despite requiring significantly more computation and memory, perform better on PathMNIST but lag behind NFGSM and AAER on earlier datasets. This illustrates that **although some baselines may match SORA in isolated settings, no existing method remains competitive across all settings**. Additionally, NFGSM and AAER exhibit lower clean accuracy even in scenarios where their robust accuracy is close to ours **(Figure 5, Right)**. In contrast, SORA is the only method that consistently achieves or surpasses state-of-the-art performance across datasets and architectures while remaining highly efficient **(Figure 6)**, demonstrating the key properties outlined in **Section 1 Lines 50–88**.
>
> # W4 - Q3 - Q4: Other attack norms, tasks, and applicability beyond $\ell_\infty$​
> Our primary focus is standard $\ell_\infty$​ image-classification benchmarks because the majority of fast-AT and CO literature uses these settings. Some fast-AT methods (e.g., ZeroGrad, MultiGrad) are tied to the $\ell_\infty$ formulation, making extensions nontrivial. Nevertheless:
>
> - The connection in **Lemma 4.1** between PertAlign and the loss Hessian holds for arbitrary perturbation directions, so PertAlign naturally extends beyond $\ell_\infty$​. We have discussed PertAlign for the special case of $\ell_2$ norm in **Corollary A.3** with some intuitions in **Remark A.4 Lines 780-782**, showing that PertAlign measures the change in direction of gradient after a small step.
>
> - We derived the optimal step for the $\ell_2$​ norm **(Proposition A.6)** and provide preliminary empirical results for $\ell_2$ in the supplement (we will include these in the final version). The extension to $\ell_1$​ is analogous.
>
> - Extending to non-image domains (text, graphs) raises domain-specific challenges (e.g., discrete tokens, graph structure). We view this as promising future work and will highlight it as such.
> | Dataset   | Architecture | Clean (%) | PGD‑10 (%) |
> |------------|---------------|-----------|-------------|
> | CIFAR‑10   | PRN‑18        | 81.94     | 53.72       |
> | CIFAR‑10   | RN‑18         | 81.50     | 53.06       |
> | CIFAR‑10   | SEN‑18        | 82.53     | 53.52       |
> | CIFAR‑100  | PRN‑18        | 56.75     | 30.19       |
> | CIFAR‑100  | RN‑18         | 56.62     | 30.18       |
> | CIFAR‑100  | SEN‑18        | 56.12     | 29.86       |

---

> ### Author Response · Authors · 2025-11-20
> **Comment 2/2: Response to Reviewer ypja (continued)**
>
> # W5 - Q5: Ablation study on hyperparameters
>
> Per the reviewer’s request, we ran hyperparameter sweeps for SORA with $\epsilon = 8/255$. Each cell corresponds to PGD-10 accuracy. Due to space limitation we have omitted the $\epsilon$ in the header for $\alpha_\max$. Results are included in the supplement and will be moved to the camera-ready version.
>
> - CIFAR10
>
> | $\beta$   | $\alpha_\max=1.5$, $\alpha_0=0.075$ | $\alpha_\max=1.5$, $\alpha_0=0.1$ | $\alpha_\max=1.5$, $\alpha_0=0.2$ | $\alpha_\max=2.0$, $\alpha_0=0.075$ | $\alpha_\max=2.0$, $\alpha_0=0.1$ | $\alpha_\max=2.0$, $\alpha_0=0.2$ | $\alpha_\max=2.5$, $\alpha_0=0.075$ | $\alpha_\max=2.5$, $\alpha_0=0.1$ | $\alpha_\max=2.5$, $\alpha_0=0.2$ |
> |--------|----------------------|-------------------|-------------------|----------------------|-------------------|-------------------|----------------------|-------------------|-------------------|
> | 0.005  | 45.53                | 45.53             | 45.53             | 48.41                | 48.41             | 48.41             | 49.54                | 49.54             | 49.54             |
> | 0.01   | 45.53                | 45.53             | 45.53             | 48.41                | 48.41             | 48.41             | 49.39                | 49.54             | 49.54             |
> | 0.05   | 45.49                | 45.63             | 45.53             | 48.37                | 48.14             | 48.50             | 49.66                | 49.43             | 49.65             |
>
> - CIFAR100
>
> | $\beta$   | $\alpha_\max=1.5$, $\alpha_0=0.075$ | $\alpha_\max=1.5$, $\alpha_0=0.1$ | $\alpha_\max=1.5$, $\alpha_0=0.2$ | $\alpha_\max=2.0$, $\alpha_0=0.075$ | $\alpha_\max=2.0$, $\alpha_0=0.1$ | $\alpha_\max=2.0$, $\alpha_0=0.2$ | $\alpha_\max=2.5$, $\alpha_0=0.075$ | $\alpha_\max=2.5$, $\alpha_0=0.1$ | $\alpha_\max=2.5$, $\alpha_0=0.2$ |
> |--------|----------------------|-------------------|-------------------|----------------------|-------------------|-------------------|----------------------|-------------------|-------------------|
> | 0.005  | 24.75                | 24.75             | 24.75             | 26.17                | 26.17             | 26.17             | 27.11                | 27.11             | 27.11             |
> | 0.01   | 24.75                | 24.75             | 24.75             | 26.17                | 26.17             | 26.17             | 27.11                | 27.11             | 27.11             |
> | 0.05   | 24.03                | 24.18             | 24.75             | 26.25                | 26.24             | 26.12             | 27.00                | 27.28             | 27.06             |
>
>
> # Q1: PertAlign vs. existing work
>
> In the fast-AT / CO context, PertAlign is specifically designed to diagnose how poorly a single-step adversary (FGSM) approximates a multi-step adversary (e.g., PGD-2). As PertAlign decreases, the discrepancy between first and second step perturbations increases **(Appendix C.3, Lines 1230–1240)**. Key differentiators:
>
> - Near-zero overhead: PertAlign reuses gradients computed for the training update and does not require extra backward/forward passes, while many alternatives (CURE/ELLE/GradAlign) add significant time/memory **(Figure 6)**.
>
> - Monitoring vs regularizing: Many prior approaches apply regularization to constrain the model (which can hurt clean accuracy). By contrast, PertAlign monitors non-linearity and adjusts the attacker adaptively, avoiding over-regularization.
>
> - Practical decision rule: PertAlign directly informs whether a second step would produce a substantially different direction; when it would, SORA reduces the step-size to avoid producing abnormal adversarial examples [6] **(Figure 2, 4)**.
>
>
> # Minor issues
>
> Thank you for the literature pointers, we will cite Robustness via Curvature Regularization (CVPR 2019) and YOPO (NeurIPS 2019) in the related work. Regarding Table 5’s caption, this was a typo: each cell contains clean and PGD-10 accuracies. We will correct this in the final version.
>
> # References
>
> [1] Intriguing properties of neural networks
>
> [2] ZeroGrad Costless conscious remedies for catastrophic overfitting in the FGSM adversarial training
>
> [3] Robustness via Curvature Regularization
>
> [4] Efficient local linearity regularization to overcome catastrophic overfitting
>
> [5] Understanding and Improving Fast Adversarial Training
>
> [6] Eliminating Catastrophic Overfitting Via Abnormal Adversarial Examples Regularization

---

> > ### Comment · Reviewer_ypja · 2025-11-25
> >
> > Post-rebuttal update:
> >
> > I sincerely thank the authors for their clarification and additional experimental results. The rebuttal has addressed part of my concerns. However, after checking comments from other reviewers and the authors' responses, I decide to keep my original rating.
> >
> > The major concerns are the lack of novelty and the marginal improvement in the experiments. I agree that the authors indeed demonstrate the relationship between nonlinear landscape and CO in a finer-grained way, but that is not enough for a top-tier conference. In addition, I agree it is challenging to consistently outperform the current strong baselines, but the numerical results do not convincingly (or clearly) demonstrate the advantages of the proposed method.

---

> > > ### Author Response · Authors · 2025-11-25
> > >
> > > We thank the reviewer for their engagement. While we understand the focus on numerical improvement on established benchmarks, we wish to clarify the nature of our contribution regarding **algorithmic stability and generalization**, which we believe distinguishes SORA from prior work.
> > >
> > > ### 1. Performance
> > >
> > > Stability and Generalization vs. Marginal Gains: We respectfully suggest that the **"marginal" improvement on saturated benchmarks (CIFAR-10) does not capture the full value of the method. In Fast Adversarial Training, stability across domains is the critical unsolved challenge.**
> > >
> > > Generalization is Non-Marginal: As shown in Figure 5, competitive baselines (e.g., NFGSM, AAER) that perform well on CIFAR effectively collapsed or suffered massive performance drops when applied to PathMNIST. In contrast, SORA maintained high robustness using a single, fixed set of hyperparameters. **The difference between a method working versus failing on a new domain is a decisive algorithmic advantage, not a marginal one.**
> > >
> > >
> > > The "Fast" AT Trade-off: We redefine the state-of-the-art by optimizing the full trade-off quartet:
> > >
> > > vs. Stronger Baselines (e.g., ELLE): While we achieve competitive robustness or even outperform them, we do so with half the training time and one-third of the memory.
> > >
> > > vs. Faster Baselines (e.g., NFGSM): While we match their speed, we provide the stability they lack on challenging datasets.
> > >
> > > Clean Accuracy: SORA consistently preserves higher clean accuracy compared to baselines with similar robustness, a crucial metric for deployment.
> > >
> > > ### 2. Novelty
> > >
> > > The First "Zero-Cost" Second-Order Metric While the relationship between curvature and CO is known, PertAlign is novel because it transforms a theoretical insight (second-order approximation) into a **zero-cost prognostic**.
> > >
> > > Prior methods (GradAlign, ELLE) require extra forward/backward passes, making them essentially "slow" Fast AT.
> > >
> > > PertAlign reuses training gradients to **predict CO before it happens**, without the computational penalty. This is a methodological advance that resolves the tension between speed and curvature awareness.
> > >
> > >
> > > SORA leverages this to adapt step sizes dynamically, solving the "Epsilon Overfitting" problem where models overfit to fixed perturbation distributions.
> > >
> > > ### Conclusion
> > >
> > > We believe that a method which **(1)** solves the brittleness of Fast AT across diverse domains (PathMNIST, ImageNet-100), **(2)** introduces a theoretically grounded zero-cost metric, and **(3)** provides a superior trade-off between efficiency and robustness, represents a substantial contribution to the field. We direct the reviewer to our [response to Reviewer PMUw](https://openreview.net/forum?id=u1lUbEDQsU&noteId=cWJQ3Ze9Qp) for evidence regarding the brittleness of baselines.

---

### Author Response · Authors · 2025-12-03
**Summary of Rebuttal and Closing Remarks**

We thank the Area Chair for their time and attention. To assist in your evaluation, we summarize our main contributions and how we have addressed the primary concerns raised during the review process.

## Core Contributions

- **Epsilon Overfitting (EO):** We identified that the non-linearity induced by Catastrophic Overfitting (CO) **follows specific characteristic patterns rather than being purely jagged** and **limited variability in perturbation distribution** significantly enhances the chances of CO. This observation motivated our use of a quadratic model to approximate loss curvature.
- **PertAlign (Zero-Cost Metric):** Leveraging this quadratic model, we proposed PertAlign, **a real-time CO predictor**. Uniquely, PertAlign incurs **zero computational overhead** by reusing training gradients.
- **SORA (Algorithmic Stability):** We introduced SORA, a second-order adaptive step-size method. **SORA matches the efficiency of the fastest first-order methods** (e.g., NFGSM) but provides superior stability and generalization.
- **Generalizability & Hyperparameter Stability:** A key insight for practical Fast AT is that methods **requiring extensive per-dataset hyperparameter tuning lose their efficiency advantage over multi-step PGD**. SORA uses a single fixed hyperparameter set across all datasets and architectures. Even with careful tuning, prior methods fail to match SORA’s performance on challenging benchmarks like PathMNIST, whereas SORA generalizes robustly without any dataset-specific adjustments.

## Resolution of Novelty Concerns
### Raised by reviewers, ypja, Xi2k, and pwii

- **Epsilon Overfitting:** Some reviews conflated EO with the known effect of large perturbation magnitude. We clarified that EO stems from limited perturbation diversity, not just size. This distinction explains why our randomization strategy effectively mitigates CO where magnitude-based theories fall short.
- **PertAlign vs. GradAlign:** We addressed the misconception that PertAlign is identical to GradAlign. We clarified three fundamental differences:

     - Mechanism: PertAlign measures curvature along the adversarial optimization direction (monitoring), whereas GradAlign measures it in a random direction (regularization).

     - Efficiency: PertAlign has zero overhead, whereas GradAlign requires costly additional forward/backward passes.

     - Theory: We derived a theoretical connection between PertAlign and the loss Hessian (Lemma 4.1), showing GradAlign is merely a special case within our broader framework.

## Resolution of "Marginal Improvement" Concerns
### Raised by reviewers, ypja, pwii, and PMUw
SORA represents a significant, non-marginal advance in Fast Adversarial Training for four key reasons:

- **Unmatched Generalization:** It is the only single-step method that consistently achieves top-tier robust accuracy across all tested datasets and architectures. Other methods either suffer Catastrophic Overfitting (0% robustness) or show severe performance drops on challenging datasets like PathMNIST.
- **True Efficiency:** Among the few methods with comparable robustness, SORA is uniquely efficient. Competitors like ELLE and MultiGrad require 2-3x more compute or memory, contradicting the "fast" premise of FAT. SORA delivers SOTA robustness without this overhead.
- **Superior Trade-Off:** When robust accuracy matches others, SORA consistently maintains higher clean accuracy (e.g., compared to NFGSM and AAER on CIFAR-10/100). It provides a better balance, avoiding the common sacrifice of clean performance for robustness.
- **Extended Validation:** In response to reviewers, we further demonstrated:
     - Generalization to $\ell_2$ adversarial training: SORA’s adaptive mechanism remains effective under the $\ell_2$ threat model.
     - Strong performance on high-resolution data: SORA outperforms prior methods on ImageNet-100, without modifying the universal hyperparameters, confirming scalability beyond low-resolution benchmarks.
     - Fair baseline comparison: We ensured a rigorous evaluation by first using all author-recommended hyperparameters for the baselines and then conducting a comprehensive grid search. This process confirmed that SORA's performance cannot be exceeded, even with extensive fine-tuning of competitor methods.

Furthermore, SORA succeeds with a single fixed hyperparameter set, while many baselines require extensive per-dataset tuning. This reliability and portability make it a genuinely practical solution, unlike methods whose fragility would force users back to slower multi-step training. The gap is substantial: SORA solves the core failures of single-step AT, Catastrophic Overfitting, poor generalization, and high cost, where others only address subsets.

We hope this summary clarifies the distinct value of our work.

---

### Meta-Review · Area_Chair_2Lic · 2026-01-05

**Summary:**

The reviewers share two main concerns: 1) the technical contribution and novelty are limited, and 2) the experimental improvements are marginal. Although the rebuttal addressed some other issues, these two most critical concerns remain unresolved.

**Reviewer Concerns:**

The reviewers share two main concerns: 1) the technical contribution and novelty are limited, and 2) the experimental improvements are marginal. These concerns are not fully addressed during the rebuttal.

**Reviewer Scores:**

Reviewer PMUw is likely to raise their score from 4 to 6. Other reviewers, however, may maintain their original scores.

---

### Decision · Program_Chairs · 2026-01-26

Reject